# Viral Shrimp Diseases Listed by the OIE: A Review

**DOI:** 10.3390/v14030585

**Published:** 2022-03-12

**Authors:** Dain Lee, Young-Bin Yu, Jae-Ho Choi, A-Hyun Jo, Su-Min Hong, Ju-Chan Kang, Jun-Hwan Kim

**Affiliations:** 1Fish Genetics and Breeding Research Center, National Institute of Fisheries Science, Geoje 53334, Korea; gene419@korea.kr; 2Department of Aquatic Life Medicine, Pukyong National University, Busan 48513, Korea; 3Department of Aquatic Life and Medical Science, Sun Moon University, Asan-si 31460, Korea; wh92dk23gus@naver.com (A.-H.J.); hongsumin0608@naver.com (S.-M.H.)

**Keywords:** shrimp disease, OIE, viral disease, DNA and RNA virus

## Abstract

Shrimp is one of the most valuable aquaculture species globally, and the most internationally traded seafood product. Consequently, shrimp aquaculture practices have received increasing attention due to their high value and levels of demand, and this has contributed to economic growth in many developing countries. The global production of shrimp reached approximately 6.5 million t in 2019 and the shrimp aquaculture industry has consequently become a large-scale operation. However, the expansion of shrimp aquaculture has also been accompanied by various disease outbreaks, leading to large losses in shrimp production. Among the diseases, there are various viral diseases which can cause serious damage when compared to bacterial and fungi-based illness. In addition, new viral diseases occur rapidly, and existing diseases can evolve into new types. To address this, the review presented here will provide information on the DNA and RNA of shrimp viral diseases that have been designated by the World Organization for Animal Health and identify the latest shrimp disease trends.

## 1. Introduction

The shrimp aquaculture industry has grown rapidly in previous decades due to increasing consumer demand, and it has consequently contributed significantly to the socio-economic development of coastal communities in many developing countries [1]. Production by the shrimp farming industry has steadily increased to approximately 3.6 million t in 2008, accounting for more than 50% of the global shrimp market, with the main production areas being in Southeast Asia, such as China, Thailand, Vietnam, Indonesia, and India, while in the Americas, the major producers are Ecuador, Mexico, and Brazil [2]. Shrimp production has steadily grown from 0.673 million t in 1990 to 6.004 million t in 2019, which is a nearly tenfold increase (Figure 1). Until recently, shrimp aquaculture production was most widespread in Latin America and East and Southeast Asian countries, but consumption is concentrated in various developed countries. Consequently, this industry is helping to reduce the economic gaps between countries by generating high levels of income in developing countries [3]. Indeed, in Southeast Asia, penaeid shrimp have contributed significantly to the economies of Indonesia, the Philippines, Vietnam, and Thailand [4].

The shrimp aquaculture industry is growing in many regions of the world, including Asia and Latin America, and it accounts for 17% of the total value of aquatic products [5]. Globally, 67% of shrimp production is from aquaculture and 33% is caught naturally, and the most common species used in shrimp aquaculture are the whiteleg shrimp, *Penaeus vannamei,* and Giant tiger prawn, the marine shrimp *Penaeus monodon*, and the freshwater prawns *Macrobrachium rosenbergii* and *Macrobrachium. nipponense* [6]. Crustacean production totaled 8.4 million t in 2017, representing an average annual increase of 9.92% since 2000, and more than 30 crustacean species were valued at 61.06 billion USD in 2017 [7]. However, with the increase in global shrimp aquaculture production, mass mortality caused by frequent disease outbreaks has become a major obstacle for the industry. Worldwide losses from disease in shrimp aquaculture in the last 15 years to 2005 were estimated to be approximately 15 billion USD, 80% of which occurred in Asia [8].

Until the 1980s, marine viruses were considered ecologically insignificant, because their concentrations were underestimated, but subsequent studies have confirmed that the ocean contains an abundance of organisms, including millions of virus particles per milliliter of seawater [9]. Most shrimp diseases are caused by viral infection, and they have an approximately four times more negative impact than bacterial diseases. In most cases, diseases caused by bacterial pathogens and parasites can be prevented through the proper management of shrimp farms (biosecurity, water quality control, stocking density, aeration, fresh feed, shrimp seed quality, and proper breeding environment), which is in contrast to viral diseases [8,10].

The occurrence of disease is the reason that existing farmed shrimp species are replaced with other species. The cause of the conversion from *P. monodon* in the 1990s to *P. vannamei* in the 2000s is also closely related to disease occurrence (Figure 2). Thailand’s *P. monodon* production increased rapidly from 1987 to the early 1990s, but thereafter, until the early 2000s, there was a large loss in production due to YHV (yellow head virus), WSSV (white spot syndrome virus), and then MSGS (monodon slow growth syndrome) [11]. Prior to 2000, *P. monodon* was the predominant aquaculture shrimp species in Asia, but the disease-free SPF (specific pathogen free) species *P. vannamei* began to increase as a replacement species (Figure 3). In Korea, the reason for the rapid replacement of *P. vannamei* from *P. chinensis*, which had been cultured since 2006, is also due to the damage caused by the frequent occurrence of WSSV (Figure 4). Ultimately, *P. vannamei* has now become the dominant shrimp aquaculture species worldwide as it is less susceptible to WSD (white spot disease) outbreaks, which had a major impact on many other shrimp species [12]. The replacement of shrimp species with *P. vannamei* in Asia has led to an increase in shrimp production from approximately 900,000 t in 2004 to 2.9 million t in 2009.

Managing the health of farmed shrimp species and developing new methods for disease prevention and treatment, preventing the illegal transboundary movement of live shrimp species, and controlling disease outbreaks through the supply of fresh food worldwide, requires an immense amount of effort. To address these issues, Flegel (2012) [8] suggested the following: (1) the development of pathogen-free SPF shrimp seeds; (2) widespread use and standardization of diagnostic tests; (3) development of biosecurity-applied breeding techniques; (4) control efforts to reduce the risk of disease transmission through cross-border movement; (5) investigations into the efficacy of immune-stimulants and vaccines; (6) a complete understanding of the specificity of shrimp species by pathogen; (7) rich epidemiologic studies of shrimp diseases; (8) molecular ecology studies to control pathogenic microorganisms in shrimp hatcheries and breeding grounds; (9) conducting virus tests through strict cross-border quarantine procedures; and (10) restricting indiscriminate imports of exotic crustaceans. This review aims to analyze the viral OIE shrimp diseases that occur frequently around the world, by examining the disease occurrence trends and diagnostic methods and providing basic data for future alternatives to shrimp diseases using the latest trend analyses and treatment plans.

## 2. DNA Viral Diseases

### 2.1. White Spot Syndrome Disease (WSSD)

Aquaculture practices are responsible for approximately 75% of the world’s shrimp production, and the predominant species used are black tiger shrimp, *P. monodon* and white Pacific shrimp, *P. vannamei* [13,14]. In the past 20 years, shrimp diseases have caused critical economic losses that seriously threaten farming practices, of which white spot syndrome (WSS) is the deadliest viral disease caused by white spot syndrome virus (WSSV) [15]. WSSV causes up to 80–100% mortality of infected shrimp within 5–10 days, thus leading to great economic loss [16]; the total economic loss from this disease is estimated to be approximately 8–15 billion USD, and this continues to increase by 1 billion USD (10% of global shrimp production) annually [17,18].

The first reports of WSD in penaeid shrimp occurred in China and Taiwan in 1992, and then spread to Korea (1993), Japan (1993), Vietnam, Thailand (1994), Malaysia (1995) and Indonesia. WSSV also occurred in America (Latin America, such as Ecuador, Mexico, and Brazil in 1999 and North America in 1995), the Middle East in 2001, and Africa (such as Mozambique and Madagascar in 2011), and most recently at an Australian shrimp farm in 2016 [19] (Figure 5). WSSV presumably reached America through the importation of *P. monodon* from Asia and became rapidly endemic in American native species such as *P. vannamei*. In Asia, during the early 2000s, the SPF species *P. vannamei* was imported from the Americas to avoid disease problems such as WSSV, resulting in the conversion of the predominant farmed species from *P. monodon* to *P. vannamei*. However, the translocation of broodstock that are unscreened or inadequately tested for WSSV has led to the spread of WSSV back to Asia from the Americas [12,19]. White spot syndrome disease (WSSD) has been listed by the World Organization for Animal Health since 1997 [20]. WSSV is considered the most serious of approximately 20 viral pathogens in shrimp, and in 2018, 46.3% of farmed crayfish in 13 provinces in China were WSSV-positive. Of note, however, is that the WSSV mortality rate in farmed crayfish is less sensitive than for shrimp, at approximately 5–90%, and it does not always lead to mortality [21].

WSSV is the only member of the genus Whispovirus in the family Nimaviridae (initially included in the family Baculoviridae) and has a double-stranded DNA genome with a virion size of 80–120 × 250–380 nm, which is rod-shaped to elliptical, and surrounded by a trilaminar envelop with a tail-like appendage [16] (Table 1). The naked viral nucleocapsid is about 80 × 350 nm and has 15 spiral and cylindrical helices of 14 spherical capsomers along its long axis, with a ‘ring’ structure at one end [22]. On the outer surface of the viral envelope, there are many tadpole-shaped spikes (5–6 nm long, 4–5 nm head diameter) to which host cells can easily attach [20]. WSSV has been reported to be approximately 300–305 kbp in length according to the isolates with 180 open reading frames (ORFs) and nine repeated sequence regions in tandem, and minisatellites (ORF 94, ORF75 and ORF125) are used for WSSV genomic and epidemiological studies [20] (Table 2). As a result, of sequencing the genes isolated from China and Taiwan, significant variations were confirmed in WSSV isolates from Vietnam and Thailand, due to the insertion of major ORF14/15 and ORF23/24 variable regions [23].

Structural proteins play important roles in cell targeting, viral entry, assembly, and budding, which is highly related to WSSV infection. Envelope protein function has a particularly critical role in viral entry to the host cell [24]. Interactions between structural proteins are common in enveloped viruses such as WSSV, but this kind of interaction involves nine WSSV virion proteins (VP19, VP24, VP26, VP28, VP37 or VP281, VP38A or VP38, VP51C or VP51, VP51A and WSV010), some of which (VP19, VP24 and VP51A) prefer self-interaction [22]. Of the envelope proteins, VP19, VP24, VP26 and VP28 are the main proteins, and VP28 and VP26 account for approximately 60% (VP28, VP26, VP24 and VP19 account for about 90%) of the envelope as the most abundant proteins [20,25]. VP28 has a critical role in the early stages of viral infection by binding WSSV to shrimp cellular receptors, and the structural protein VP24 is a key protein that directly binds to VP26, VP28, VP38A, VP51A, and WSV010 to form a membrane-associated protein complex [22]. WSSV VP28 is an adhesion protein that helps the virus to bind to shrimp cells and enter the cytoplasm during infection, and VP26 may bind to actin or actin-related proteins and help WSSV translocate to the nucleus [9]. In addition to VP28, VP37 is a viral envelope protein known to promote WSSV infection through binding to shrimp cells, resulting in virus binding to the hemocytes [26]. Furthermore, structural proteins of the virion envelope such as VP26, VP31, VP37, VP90, and VP136 interact with integrin receptors to stimulate the binding of viruses to the extracellular matrix (or intercellular adhesion) [13,27].

WSSV isolates from several regions with different genotypes [Thailand (GenBank no. AF369029), Taiwan (GenBank no. AF440570), China (GenBank no. AF332093), and South Korea (GenBank no. JX515788)] have been sequenced, but they are all classified as a single species of the genus Whispovirus (family Nimaviridae) [24,28]. The complete genome sequence of WSSV isolates was reported in 2001 (WSSV-TH, GenBank no. AF369029; WSSV-CN, GenBank no. AF332093), 2002 (WSSV-TW, GenBank no. AF440570), 2013 (WSSV-KR, GenBank no. JX515788), 2016 (WSSV-MX08, GenBank no. KU216744), 2017 (WSSV-CN02, CN01 and CN03, GenBank no. KT995470-995472; WSSV-CN04, GenBank no., KY827813; WSSV-CN-Pc, GenBank no. KX686117) and 2018 (WSSV-AU, GenBank no. MF768985; IN_AP4RU, GenBank no. MG702567; WSSV-EC-15098, GenBank no. MH090824; WSSV-chimera, GenBank no. MG264599) and 2020 (CN_95_DFPE, GenBank no. MN840357) [29,30,31,32,33,34,35,36,37,38,39,40,41] (Table 3). The major deletion region at ORF23/24, variable region at ORF14/15, and variable number tandem repeats (VNTRs) located in ORF75, ORF94, and ORF125 are used as genetic marker to differentiate WSSV genotypes [23,36,42]. Mx-F, Mx-H, Mx-C, and Mx-G strains (GenBank no. HQ257380-257383) have 99–100% identity to each other in the ORF14/15 region and all four contain a 314 bp region present only in isolated WSSV-In-07-I (GenBank no. EF468499). The low-virulence strain Mx-G has additional repeat units (RUs) in ORF94 when compared to the highly virulent strain Mx-H, and both have 100% identity in the variable number of tandem repeats (VNTR) in ORF75 and ORF125 [28]. During the spread of WSSV in Asia, significant changes were observed in the ORF14/15 and ORF23/24 regions, and consequently, WSSV strains increased host mortality, shortened host survival, and developed increased competencies in host competition [43].

WSSV is known to be highly pathogenic to crabs, copepods, and other arthropods, including penaeid shrimp (*P. monodon*, *P. indicus*, *P. japonicus*, *P. chinensis*, *P. penicillatus*, *P. semisulcatus*, *P. aztecus*, *P. vannamei*, *P. merguiensis*, *P. duorarum*, *P. stylirostris*, *Trachypenaeus curvirostris,* and *Metapenaeus ensis*), caridean shrimp (*Exopalaemon orientalis* and *M. rosenbergii*) and crayfish, *Procambarus clarkii* [44] (Table 4). Of the more than 100 potential host species for WSSV, it is particularly lethal to all marine aquaculture shrimp which are more vulnerable to WSSV than freshwater shrimp and other species, even though the susceptibility of a potential host to WSSV may vary from species to species [20,45]. During all stages of development, from egg to adult, species are vulnerable to WSSV [46].

Shrimp infected with WSSV are characterized by anorexia, lethargy, abnormal behavior (decreased swimming ability, disorientation and swimming on one side), red discoloration of the body surface (uropods, telson, pereiopods, and pleopods), swelling of branchiostegites, a loosening of the cuticle, enlargement and yellowish discoloration of the hepatopancreas, thinning and delayed clotting of hemolymph, and characteristic white spots with a diameter of 1–2 mm (or 0.5–3.0 mm) on the carapace, appendages, and internal surfaces during disease progression [47] (Figure 6). WSSV infection in shrimp is easily recognized by the characteristic white spots on the carapace, but WSSV infection does not always show symptoms of white spots and cannot be considered as a reliable indication for the diagnosis of disease, as some bacterial infections, high alkalinity, and stress can also produce similar spots [48]. Although the exact mechanism of white spot formation by WSSV infection is not known, WSSV infection can cause integumentary dysfunction, resulting in accumulation of calcium salts in the cuticle, resulting in white spot formation [49]. WSSV proliferates in the nucleus of the target cell in the subcuticular epithelium, gills, lymphoid organs, antennal glands, hematopoietic tissue, connective tissue, ovaries, and ventral nerve cord. In the later stages of infection, the infected cell is degraded and the tissue destroyed [50].

WSSV replicates rapidly in the host’s cells after infecting the host, and usually causes host death within one week [51]. WSSV frequency can be influenced by a variety of environmental stressors, such as temperature changes, salinity reductions, and pH fluctuations [27]. The transmission of WSSV disease can occur through the feeding of infected individuals, and horizontal transmission through the water-borne route has also been demonstrated. Individuals surviving WSSV infection can carry the virus for life and transmit it to their offspring through vertical transmission via oocytes [52]. Aquatic and benthic organisms such as polychaete worms, microalgae, and rotifer eggs are known vectors of WSSV, and 43 arthropods have been reported as hosts and vectors of WSSV in culture facilities, aquatic systems, and experiment [18]. Shrimp infected with WSSV usually congregate near the edge of the pond and show clinical signs one to two days before death occurs [20]. WSSV disease susceptibility in crabs, crayfish, freshwater prawns, spiny lobster, and clawed lobsters is highly variable, but in penaeid shrimp, the cumulative mortality rate is typically 90–100%, 3–10 days post-infection and WSSV is fatal to penaeid shrimp [18]. WSSV usually shows clinical signs in farmed penaeid shrimp at 14–40 days and shows a high mortality rate with up to 100% mortality in sensitive hosts.

WSSV diagnostic technology is evolving from the previous, morphology-based identification to more highly sensitive immunological and molecular technologies that can detect viruses, even in asymptomatic carriers, using electron microscopy (EM) [53]. Among various diagnostic methods, PCR is used as the most sensitive method by which to detect WSSV infection, by targeting the VP28 gene [27] (Table 5). There are several PCR methods available for the diagnosis of WSSV, such as one-step PCR, nested-PCR, and real-time PCR [54]. One-step PCR can be used to detect the presence of WSSV in shrimp with high levels of infection, and nested-PCR can increase the sensitivity level when compared to one-step [55] to detect low levels of infection in the broodstock, nauplii, post-larvae, and juvenile stages [54]. Therefore, the pathogen can be easily detected using one-step PCR when clinical signs such as lethargy, reduced feeding and white spots on the exoskeleton appear, but can only be detected by nested-PCR when asymptomatic [55]. In addition, real-time PCR is a reliable technique by which to monitor the entire analysis in actual time through the detection and quantification of WSSV virion copy number [27]. Hematoxylin and eosin (H & E) histology is an important diagnosis method that is used to verify WSSV infection in shrimp [56]. Histological diagnosis following WSSV infection occurs in all tissues of mesodermal and ectodermal origin such as gills, lymphoid organ, cuticular epithelium, and sub-cuticular connective tissues, and infected nuclei are enlarged with alienated chromatin and contain inclusion bodies with strongly stained eosinophils in early infection and basophils in more advanced infections [18] (Figure 7). Biosecurity measures (specific pathogen-free (SPF) broodstock, complete dry-out of culture tanks after harvest, low water exchange systems such as RAS), restricting access to vectors and pathogens (through crab fence, bird blocking, and foot baths in shrimp farm entrance), and improving disease resistance (immunostimulants, neutralization, environmental management and vaccines) in shrimp are effective management methods, as there is currently no way to treat WSSV infection [20].

### 2.2. Infectious Hypodermal and Hematopoietic Necrosis Virus (IHHNV)

Infectious hypodermal and hematopoietic necrosis virus (IHHNV) is a critical viral pathogen of penaeid shrimp, causing serious economic loss to the shrimp aquaculture industry (up to 50% of the overall economic loss in shrimp aquaculture), and has been listed as a reportable crustacean disease pathogen by the World Organisation for Animal Health (OIE) since the year of 1995 [57]. IHHNV was first detected in blue shrimp, *P. stylirostris* post-larvae and juvenile imported from Costa Rica and Ecuador at a shrimp farm of Hawaii in 1981, causing up to 90% mortality, and it was discovered in the quarantine process of imported white leg shrimp, *P. vannamei* at a shrimp farming facility in Taiwan in 1986, and in giant tiger prawn, *P. monodon* aquaculture of Australia in 2008 [58].

Since IHHNV was first reported in blue shrimp, *P. stylirostris*, IHHNV disease outbreaks had been reported in more than 20 countries in Asia, America, Africa and Oceania, such as Korea, Philippines, Singapore, Malaysia, Thailand, Indonesia, USA, Brazil, Mexico, Argentina, India, Venezuela, Mozambique, Madagascar, Tanzania and Australia [59] (Figure 8). IHHNV infects the major aquaculture shrimp species, *P. stylirostris* and *P. vannamei*, in North America, which is causing economic losses [60]. IHHNV is lethal in juvenile *P. stylirostris* with 90% mortality (acute disease), whereas it causes runt deformity syndrome (RDS; asymptomatic carrier of the virus) in *P. monodon* and *P. vannamei*, reducing the market value by 10–50% [61]. IHHNV causes the RDS in juvenile *P. vannamei* and *P. monodon*, which causes stunting in growth, and accounts for 50% of the economic loss in the shrimp industry [59,62] (Figure 9). IHHNV causes economic damage by reducing the marketability of shrimp due to poor growth, irregular growth, and epidermal malformation during harvest by RDS (cuticular deformities of the rostrum, antennae, thoracic and abdominal areas) [63,64] (Figure 10).

IHHNV is a linear single stranded DNA virus of 3.9 kb in length and the smallest penaeid shrimp virus that is non-enveloped and icosahedral linear virion with an average diameter of 22–23 nm [60]. IHHNV was taxonomically a *Penaeus stylirostris* densovirus (PstDNV) from the Parvoviridae family, Densovirinae subfamily, but in July 2019, ICTV (International Committee on Taxonomy of Viruses) reconstituted the Parvoviriae family as the Parvoviridae family, Hamaparvovirinae subfamily, and Penstylhamaparvovirus [59]. IHHNV has a capsid made up of four polypeptides with molecular weights of 74 k, 47 k, 39 k and 37.5 k [65]. IHHNV may exhibit different virulence due to differences in genotype of IHHNV, host susceptibility and developmental stage of infected shrimp; (i) Acute infection: IHHNV-infected post-larvae and juveniles *P. stylirostris* sink to the bottom without swimming and can cause up to 90% of shrimp mortality in a short period of time; (ii) Chronic infection: Mass mortality does not usually occur in IHHNV-infected juvenile *P. vannamei* and *P. monodon*, and sub-adults *M. rosenbergii*, which can cause RDS such as growth and rostrum retardation, abdominal and tail fan deformation, cuticular roughness, and wrinkled antennal flagella, resulting in 30–90% growth retardation; (iii) Asymptomatic carriers: *Mytilus edulis* and adult *M. rosenbergii* can carry the infectious IHHNV type, but do not show major clinical and pathological symptoms and serve only as carriers; (iv) non-infectious IHHNV insertion into shrimp host genome: Exposure to IHHNV was not infectious in *P. monodon* and *P. vannamei* individuals injected with crude extracts of *P. monodon* carrying the IHHNV sequence through feeding and injection [59].

Genetic characterization of multiple IHHNV strains isolated from multiple regions can determine whether the virus has evolved or not and the existence of other strains in the region with exogenous sources [58] (Table 3). The IHHNV genome consists of three ORFs (open reading frames): two encoding nonstructural proteins (NS1; 2001 bp and NS2; 1092 pb) and one encoding viral capsid proteins (CP; 990 bp) [57,59] (Table 2). Of five genotypes classified in IHHNV, type I, type II, and type III are infectious types, and type A and type B are non-infectious. Type I was found in *P. monodon* of Australia (GenBank no. CQ475529.1); type II was mainly found in the United States and Southeast Asia (GenBank no. AY102034.1, JN616415.1, AY362547.1, etc.), and type III was mainly distributed in East Asia (GenBank no. AY355308.1, EF633688.1, KF214742.1, and JX258653.1, etc.) [59] (Table 3). Two IHHNV virus sequences were found in *P. monodon* in Africa (Type A was found in Madagascar and Australia, and type B was found in Tanzania). Type A and type B sequences have three ORFs with high similarity, which has the identical replication initiator motif and NTP-binding and helicase domains with IHHNV virus, but both type A and type B IHHNV-related sequences are non-infectious genotypes [66].

IHHNV was found in *P. monodon* in Southeast Asia (Thailand, Taiwan, and the Philippines), and only about 30 animal species are known to be IHHNV-susceptible or carriers of IHHNV [59]. IHHNV mainly affects Penaeid shrimp, but *Artemesia longinaris*, *Palaemon macrodactylus* and post-larvae and subadults of *M. rosenbergii* as well as *P. clarkii* are also known to be naturally infected with IHHNV. Bivalve shellfish and adults of *M. rosenbergii* act as carriers in IHHNV without infection-related symptoms [57]. For example, in the IHHNV PCR test on the coast of China, the positive rate of IHHNV in the gills, muscles and gonads of *Mytilus edulis* was more than 80%, but the pathogenicity of IHHNV infection was not shown. In addition, the pathogenicity of IHHNV infection was closely related to the age and size of the host, and in general, young shrimp are more susceptible to IHHNV infection [59]. Larval and juvenile *P. stylirostris* at 0.05–2 g is more susceptible to IHHNV, especially *P. stylirostris* at 0.08 g is most susceptible to IHHNV, whereas *P. stylirostris* at 2 g or more significantly weakens IHHNV pathogenicity. Adults of *M. rogenbergii* do not show obvious symptoms of IHHNV infection, but IHHNV infection in subadults can cause slow growth and cause RDS also in juvenile of *P. vannamei* and *P. monodon*, whereas adult *P. vannamei* showed no obvious pathological symptoms [62]. IHHNV shows a marked difference in pathogenicity according to the infecting shrimp species; While *P. sylirostris* is highly pathogenic, *P. vannamei* causes RDS, a chronic disease [67].

Because IHHNV does not encode a DNA polymerase and is dependent on the host cell for DNA replication and proliferation, it requires the host’s rapidly proliferating cells for replication; the main target organs for IHHNV infection contains tissues of ectodermal (cuticular epidermis, nerve cord and ganglia, hypodermal epithelium of the fore and hind gut) and mesodermal (antennal gland, lymphoid organ, hematopoietic organs, striated muscles, tubule epithelium and connective tissue) origin, but IHHNV does not affect tissues of endodermal origin such as hepatopancreas, anterior mid-gut caecum, midgut epithelium or posterior midgut caecum [58] (Table 4). It is the post-larvae and juvenile shrimp that are susceptible to IHHNV owing to the reason that they have actively dividing cells. The *P. stylirostris* presents acute symptoms of IHHNV such as white or buff-colored spots at the junction of the tergal plates in the abdomen, whereas IHHNV in the *P. vannamei* appears as a chronic disease, RDS, showing symptoms such as wrinkled antennal flagella, ‘bubble-heads’, deformed rostrum, cuticular roughness and deformation in 6th abdominal segment and tail fan [59].

Shellfish, as an important carrier of IHHNV disease, have a very high risk of transmission, but the mechanisms of infection and pathogenicity are still unclear in many respects [59]. In the case of horizontal transfer of infection, the *P. sytlirostris* surviving IHHNV infection can become life-long carriers of the virus and cause spread through vertical and horizontal propagation. In the natural environment, IHHNV transmission can occur horizontally through shrimp feeding and water, and vertical transmission can occur from mother to offspring [58]. IHHNV was detected in the ovaries of IHHNV-infected females, whereas the IHHNV did not appear in the sperm of infected males, so vertical transmission of IHHNV from infected females was clearly established [67]. Post-larvae *M. rosenbergii* with IHHNV infection showed a high mortality rate of up to 80–100, and juvenile and subadult *P. stylirostris* showed a mortality rate of up to 90% (however, *P. stylirostris* also has increased resistance to IHHNV infection, and no significant mortality has recently been reported.); on the other hand, in *P. vannamei* and *P. monodon*, IHHNV was less virulent with no death, just including RDS such as stunting and cuticular deformities [58,66].

In an epidemiological survey, the IHHNV prevalence of shrimp in aquaculture areas was 51.5% and 8.3% for shrimp and crab in China, 9.4~81% for shrimp in northeastern Brazil, 14.1%for *P. monodon* in Brunei Barussalam and 30% for *Artemesia longinaris* in Argentina, 1.1~3.3% for *P. vannamei* in Venezuela, 20% for *M. rosenbergii* in Malaysia [58]. Currently, the most reliable techniques used for IHHNV detection are conventional PCR and real-time PCR. However, since the existing PCR cannot quantify the virus in the infected sample, the real-time PCR technique (probe-based and dye-based methods) is more useful [68] (Table 5). TaqMan probe-based real-time PCR is also a sensitive technique for IHHNV detection (Table 5). Encinas-García et al. (2015) [69] developed SYBR Green-based real-time PCR for the detection and quantification of IHHNV in *P. sylirostris*, which is much cheaper and simpler than TaqMan probe real-time PCR (Table 5). Histologically, the diagnosis of IHHNV infection is made through the identification of prominent Cowdry type A, eosinophilic, intra-nuclear inclusion bodies enclosed by marginated chromatin in hypertrophied nuclei of cells in tissues of ectodermal and mesodermal origin [58]. In electron microscopy of negatively stained IHHNV VLPs in *P. vannamei*, IHHNV-VLPs were uniformly spherical and 23 ± 3 nm in diameter, similar to native IHHNV particles [70] (Figure 11A). H&E staining of *P. monodon* infected with IHHNV showed intra-nuclear Cowdry type A eosinophilic inclusion bodies [64] (Figure 11B). Several hypertrophied nuclei were observed in the gill tissues of IHHNV-infected *P. clarkii* [71] (Figure 11D). An effective vaccination strategy for IHHNV has not been developed, and there are no confirmed reports of effective chemotherapy and immune-stimulation treatment [72]. As there is currently no effective treatment for IHHNV, the best management strategy is to screen SPF shrimp for IHHNV, but when IHHNV cannot be completely controlled, IHHNV-resistant shrimp populations may be used.

## 3. RNA Viral Diseases

### 3.1. Infectious Myonecrosis Virus (IMNV)

Infectious myonecrosis (IMN), also known as Penaeid shrimp myonecrosis virus (PsIMNV), is a major disease caused by the infectious myonecrosis virus (IMNV), which adversely affects the shrimp aquaculture industry [73,74]. IMN was first identified in Piaui state, Brazil in August 2002, and then rapidly spread through the coastal areas of northeastern Brazil, which significantly reduced the productivity of the Brazilian shrimp aquaculture industry in 2004 and 2005 [75]. In the Asia-Pacific region, *P. vannamei* is steadily increasing in importance as a major aquaculture species. Furthermore, IMNV was added to the World Organization for Animal Health in 2005 and NACA (Network of Aquaculture Centres in Asia-Pacific)/FAO (Food and Agriculture Organisation) in January 2006 due to large-scale transboundary movements of the disease and its impacts on aquaculture species [62,76]. In Brazil this pathogen caused an economic loss of approximately 20 million USD with 40–60% mortality in 2003. By the end of 2005 the economic losses as a result of the IMNV outbreak had reached 430 million USD, and by the end of 2011, Brazil and Indonesia had suffered a combined economic loss of approximately 1 billion USD in Brazil and Indonesia [76,77].

IMNV was first reported in 2003 in *P. vannamei* cultured in northeastern Brazil, then in Indonesia (2006), and most recently in India (2016), Malaysia (2018) and Indonesia (2018) [78,79] (Figure 12). Until the IMNV virus was reported in India in 2016, it had only occurred in Brazil and Indonesia [80]. IMNV occurs in *P. vannamei,* its infectious host, and causes infective myonecrosis. The occurrence of this disease is thought to be related to certain types of environmental and physical stress (extreme temperature and salinity, collection by cast-net) and the use of low-quality shrimp feed [62]. Although IMNV can induce an increase in mortality due to an acute infection in *P. vannamei*, the infection is usually detected by observing chronic symptoms in the host rather than a rapid mortality. The symptoms displayed by *P. vannamei* infected with IMNV include focal to extensive white necrotic areas in the striated muscle, especially the distal abdominal segments and tail fan [79], as well as a slow mortality that persists during the culture period (cumulative mortality reaching up to 70%) [81].

IMNV is a single molecule of double-stranded RNA forming a monopartite genome that is 7561–8230 bp in length with two open reading frames (ORFs). It is a non-enveloped icosahedral virus with a diameter of 40 nm and fiber-like protrusions on the surface [74,82] (Table 1). IMNV is taxonomically a totivirus belonging to Totiviridae family that is similar to Protozoa and Fungal viruses. In a phylogenetic analysis based on RdRp, IMNV was identified as a member of the Totiviridae family in 2008 [74,83]. The Totiviridae family consists of five genera (Giardiavirus, Leishmaniavirus and Trichomonasvirus, which infect protozoa; and Totivirus, and Victorivirus, which infect fungi) recognized by the ICTV (International Committee on Taxonomy of Viruses), but many researchers have recently suggested that the Arthropod Totiviruses should be classified separately as an Artivirus genus within the Totiviridae family [76].

Whole-genome sequencing of IMNV revealed two ORFs such as ORF1, encoding RNA binding and capsid proteins and ORF2, encoding putative RNA-dependent RNA polymerase (RdRp) [83] (Table 2). The coding region of the RNA-binding protein is situated in the first half of ORF 1 (including a dsRNA-binding motif). The second half of ORF1 encodes a capsid protein with a molecular mass of 106 kDa [77]. The function of the dsRBM (dsRNA binding motif) is critical for modulation and viral replication in the immune system of the shrimp host. However, the functions of small proteins are still unclear, but hypotheses have been suggested in which they may be connected to assembly, cell entry, and extracellular transmission of the virus [76]. ORF2 demonstrates high similarity to the RdRp of the Totiviridae family, and ORF2 coding strategies of IMNV are similar to the strategies of GLV (Giardia lamblia virus) and other members of the Totiviridae family, which indicates that RdRp is a conserved domain [76].

IMNV strains identified in Brazil (six strains) and Indonesia (ten strains) showed high similarity with the alignment of a 372 bp fragment encoding the major capsid protein (MCP) of IMNV strains isolated from the two regions. This suggests that the MCP could be used as a target gene to track the movement of IMNV [77] (Table 3). Through subsequent analysis, it was confirmed that the IMNV in Brazil and Indonesian reported by GenBank had nucleic acid sequence identity of 99.6% [82]. The capsid protein has a major role in virus adhesion, virulence, and cell entry, and the MCP gene (nt 2248~4953) of IMNV also contains a variable region with 72 polymorphic sites, so that the MCP gene sequence can be used to trace the origin of a new strain [82].

IMNV not only infects *P. vannamei*, which are naturally susceptible to it, but also *P. stylirostris* and *P. monodon*, which have been found to be experimentally susceptible. Furthermore, the wild Southern brown shrimp, *Penaeus* (*Farfantepenaeus*) *subtilis* is also susceptible to IMNV infection [76,84]. IMNV is known to only infect Penaeid shrimp (4 shrimp species: *P. vannamei*, *P. sylirostris*, *P. monodon*, *P. subtiltis*), but can do so at all life stages including post larvae, juvenile, and adult, but mortality was observed only in the juveniles and adults showing symptoms of a cooked appearance [76,79] (Figure 13). In IMNV-infected shrimp, extensive white necrosis of the striated muscle, especially the distal part of the abdomen and tail fan, may progress, and dissection of moribund shrimp may show enlarged lymphoid organs more than twice the normal size [62] (Figure 13C,D). Clinical signs of IMNV are prominent in the acute phase of infection, and although the main target organ is the skeletal muscle, gills and lymphatic organs may also be affected. IMNV infection in the chronic stage can be identified by necrotic muscle liquefaction exhibiting coagulative muscle necrosis [76]. Typical symptoms of IMNV infection include transparency loss, abdominal and cephalothorax necrosis, tail coloration, hepatopancreas volume loss, and progressive tail fan necrosis [85]. Shrimp infected with IMNV are characterized by whitish or reddish discolorations in the tail muscle and opaque, whitish discolorations in the abdominal muscle due to white necrosis in the striated muscle [86]. Coelho et al. (2009) [75] suggest that shrimp infected with IMNV lose transparency, and this symptom starts at around the second or third segment and then extends towards the telson.

The first report of IMNV occurred in a shrimp farm in northeastern Brazil in 2002 and it then spread to a shrimp farm in Indonesia in 2006. The cause of IMNV transmission is believed to be the uncontrolled movement of brood stocks and post larvae shrimp across borders [87]. Since it was first reported from Brazil, the origin of IMNV is thought to be South America, and the geographical distribution of the disease is limited. Although the exact mechanism for IMNV transmission is unknown, there is also the possibility of horizontal transmission through cannibalistic behavior or the water via infected shrimp, and vertical transmission from broodstock to progeny [76]. The source of vertical transmission is assumed to be maternal based on the low sperm cell survival rate of naturally infected males and the 100% positive occurrence in the ovaries of female shrimp infected with IMNV [82]. Although specific data on the vector of IMNV are lacking, it has a non-envelope particle structure like TSV (non-enveloped virus particles have high survival rates in the gastrointestinal tracts of animals), and thus has the potential to maintain infectivity in the intestines and feces of seabirds that feed on IMNV-infected dead or dying shrimp [88].

IMNV infection progresses slowly throughout the growing season with low mortality, but cumulative shrimp mortality in ponds during harvest can reach up to 70% [86]. In general, the mortality rate due to IMNV infection is between 20–50%, and the mortality rate gradually increases, resulting in 40–70% mortality during the growing season [83]. Given that the major target tissues of IMNV are the striated skeletal muscles which are not considered vital tissue, the virulence following IMNV infection is less lethal, when compared to other viruses such as WSSV, YHV, and TSV. In addition, the damage at the early steps of IMNV infection can be repaired in the muscle tissues [76]. Although IMNV is not fatal when compared to WSSV and YHV, this virus is a stress-dependent virus, which is lethal to *P. vannamei* when there are rapid changes in water quality parameters such as pH, temperature, plankton, and dissolved oxygen [82]. Due to its slow disease progression, IMNV can cause significant economic losses due to high feed conversion efficiency as the infected individuals consume feed continuously [76].

IMNV infection is diagnosed primarily through clinical symptoms, histopathological examination, and molecular techniques [74]. Since there are no effective drugs or vaccines available for IMNV, a sensitive and reliable diagnosis is required for appropriate control measures. The TaqMan real-time RT-PCR assay provides a rapid and sensitive method for clinical diagnosis of IMNV [89] (Table 5). Histological lesions due to IMNV infection are characterized by coagulative myonecrosis, with hemocytic infiltration, fibrosis, and fluid accumulation in muscle fiber (edema) [90] (Figure 14). Among shrimp challenged with IMNV, 10% showed a light coagulation and hemocyte infiltration [75]. During the acute phase of IMNV, the main target organs are the striated muscles, hemocytes, connective tissues, and lymphoid organ tubule parenchyma cells, whereas the major tissues targeted during the chronic phase are the lymphoid organs [76]. During the acute or chronic phase of IMNV, considerable hypertrophy of the lymphoid organs, induced by the accumulation of lymphoid organ spheroids (LOS), results in the development of consistent lesions [62].

As there is currently no effective method by which to control the spread of or treat IMNV, prevention, management, and prompt diagnosis are the most effective tools [87]. Experimental infections showed that here was 20% mortality in *P. vannamei*, but 0% mortality in *P. stylirostris* and *P. monodon*. Therefore, restocking with IMNV-resistant individuals such as *P. monodon* and *P. stylirostris* could be a useful method to reduce mortality losses [76]. To prevent the vertical transmission of IMNV, eggs and larvae must be disinfected, and biological security measures, appropriate quarantine, and SPF (specific pathogen free) bloodstocks procedures implemented, in addition to stocking density decreases, stress reduction in the culture environment, and immune-stimulant administration [82].

### 3.2. Yellow Head Virus Genotype 1 (YHV Genotype 1)

Yellow head virus (YHV-1) and gill-associated virus (YHV-2; GAV) first emerged in the early to mid-1990s and are serious pathogens of the giant tiger shrimp, *P. monodon* farmed in Thailand and Australia, respectively [91]. Although YHV-1 and YHV-2 (GAV) share the same susceptible host, *P. monodon*, they have geographically distant natural distributions and show significant differences in virulence and pathogenicity [92]. Of the eight identified genotypes, typical symptoms of YHV infection in shrimp are known only for the YHV genotype 1 [93], and losses due to YHV were estimated to be between 30 to 40 million USD in Thailand in 1995, before the outbreak of WSSV [94].

The YHV genotype 1 is the most virulent, was first identified in *P. monodon* cultured in Thailand in 1990 [95] (Figure 15), and it caused mass mortality of the species and significant economic losses to the shrimp industry. It was designated as a notifiable disease by the World Organisation for Animal Health (OIE) in 1995 [68]. It was first observed in cultured black tiger shrimp, *P. monodon* in central Thailand in 1990, and by 1992 had spread to shrimp farming areas on the eastern and western coasts of the Gulf of Thailand. In 1993, a virus morphologically identical to YHV genotype 1 was detected in the lymphoid organs of healthy wild and farmed *P. monodon* in Queensland, Australia, and was thereafter named the lymphoid organ virus (LOV). YHV was then detected at high levels in gills with YHD (yellow head disease)-like histopathology in the gills of moribund aquaculture *P. monodon* between 1995 and 1996 and was named GAV (gill-associated virus) [95].

There have been reports of YHD infection in farmed *P. vannamei* and *P. stylirostris* in Mexico, but it has not been confirmed, and there are no official reports of YHV infection in the Americas [96]. YHD has also been reported in *P. monodon* in Asian countries such as Vietnam, Philippines, Sri Lanka, Indonesia, Malaysia, India, and China, but has rarely been confirmed by laboratory analysis [97]. GAV, a YHV strain in Australia (YHV genotype 2), is related to a disease called mid-crop mortality syndrome (MCMS) in *P. monodon* in Australia, which was also detected in black tiger shrimp, *P. monodon* farmed in Vietnam and Thailand [98]. GAV is a chronic infection in Australia, causing significant economic losses to the Australian shrimp aquaculture industry since 1996, and GAV infections have been reported in farmed and wild *P. monodon* along the eastern coast of Australia [99]. YHV genotype 3 was detected in Taiwan, Vietnam, Indonesia, Malaysia, Thailand, and Mozambique, and YHV genotype 4 was found in India, which is the most frequently detected genotype. YHV genotype 5 was detected in the Philippines, Malaysia, and Thailand, and YHV genotype 6 was detected in Mozambique [100]. YHV genotype 7 was detected in *P. monodon* infected with the disease in Australia in 2012 [101]. In China, YHV genotype 1 was first detected in *P. monodon* imported from Thailand by the Shanghai Entry-Exit Inspection and Quarantine Bureau in 2005, and a new genotype YHV 8 was discovered in Hebei, China in July of 2012 [68].

YHV genotype 1 is a positive sense, rod-shaped, enveloped single-stranded RNA genome with virions of 40–60 nm × 150–200 nm and internal helical nucleocapsids of 15 nm in diameter 80–450 nm in length [94,100]. YHV is taxonomically classified in the Okavirus genus belonging to the Roniviridae family within the Nidovirales order [102] (Table 1). The virions of YHV include a polyadenylated 26.6 kDa genome and three structural proteins with transmembrane glycoproteins gp64 and gp116, the components on the virion surface [100]. YHV virions include three structural proteins, such as two transmembrane glycoproteins (gp116 and gp64) and a nucleoprotein (p20), and the envelope glycoprotein (gp116) has been shown to be the main virulence factor of YHV genotype 1 [103]. The genotypes that have evolved from *P. monodon* individuals are geographically separated from YHV and have evolved into YHV (YHV genotype 1) and GAV (YHV genotype 2) forms, which are indistinguishable [91].

The genome includes five canonical long ORFs (ORF1a, ORF1b, ORF2, ORF3, and ORF4), in order from the 5′-end: encoding replicase enzymes (ORF1a, overlapping ORF1b); encoding the nucleoprotein, p20 (ORF2); encoding the precursor polyprotein, pp3 that is processed to produce envelope glycoproteins such as gp116 and gp64 (ORF3) [104] (Table 2). YHV (YHV genotype 1) and GAV (YHV genotype 2) share a similar genome as the level of nucleotide sequence identity between them is approximately 79% overall (approximately 74% for ORF3 and 82% for ORF1b); the level of amino acid sequence identity between the genomes is 73% for gp116 and 84% for pp1ab [92]. The YHV genome (26,662 nt) is larger than the GAV genome (26,235 nt) owing to the sequence insertions occurring in several large blocks, whereas the GAV genome has few sequence insertions [92]. After YHV was first reported in Thailand in 1990, eight geographic types of genotypes have been reported, with genotypes differing by up to 20% in virulence and whole genome sequence [105] (Table 3). The mutant YHV genotype was also detected in healthy *P. monodon* broodstock in Thailand and was reported in *P. monodon* and *P. japonicus* which were cultured in Taiwan [97]. YHV genotype 1, the only virulence genotype of YHV was first reported in 1990 with typical signs of yellow head disease, which caused the mass mortality of *P. monodon* in Thailand [68]. YHV genotype 2 (GAV) is the only disease-associated YHV gene line other than YHV genotype 1 and is associated with a less severe form of the disease in Australian farmed shrimp [98]. Senapin et al. (2010) [106] suggests that GAV induces MCMS, which have lower virulence levels than those for YHV genotype 1 which is 106 times more virulent.

Most aquacultured species of penaeid shrimp, including *P. stylirostris*, *P. aztecus*, *P. duorarum*, *P. setiferus,* and *P. vannamei*, are susceptible to YHV-1 infection, while *P. esculentus*, *P. merguiensis,* and *P. japonicus* are susceptible to GAV [107] (Table 4). YHV infection also caused high mortality in *Marsupenaeus japonicus*, *P. vannamei*, *P. stylirostris*, *P. esculentus*, *P. merguiensis*, *P. setiferus*, *P. aztecus*, *P. duorarum*, *M. ensis,* and *M. affinis* [100], but *P. monodon* was the most affected overall [108]. It was observed that juvenile and sub-adult shrimp are susceptible to YHD and mortality within a few hours after showing clinical symptoms [95]. The GAV and YHV genotypes (YHV 3~8) have also been reported in healthy *P. monodon* from Indonesia, Malaysia, the Philippines, Vietnam, Thailand, Taiwan, Brunei, India, Mozambique, and Fiji [100].

YHV genotype 1 infection presents typical disease symptoms with yellow coloration of the cephalothorax and gills, but YHV-1 infection can exist for long periods without any signs of disease, such as with the WSSV outbreaks [102]. Samocha (2019) [109] also reported yellow discoloration of the cephalothorax and gills of *P. monodon* infected with YHV-1 (Figure 16). YHV-1 infection faded the overall body color of the shrimp, and mortality progressed after about 45–60 days of culture, resulting in a cumulative mortality rate of 60–70% [106]. Prapavorarat et al. (2010) [110] reported that after the initial clinical signs of YHV-1 disease (the development of yellow discoloration of the cephalothorax and gills), 100% mortality occurred within 3–9 days, resulting in rapid damage to shrimp production. As a result, of dissecting moribund shrimp due to YHV-1 infection, hepatopancreatic atrophy was reported [68]. YHV-1 affects tissues of ectodermal and mesodermal origin, and leads to critical lymphoid organ and gills necrosis [1]. In acute GAV infection, yellow cephalothorax lesions were not clearly seen, and general redness of the body and gills was observed, which was reproduced in artificial GAV challenge infection experiments in the laboratory [95]. GAV is very prevalent in penaeid shrimp and does not cause disease in healthy shrimp, other than a chronic infection [99]. Acute infection with YHV-1 and GAV can affect all mesodermal and ectodermal tissues containing lymphoid organs, circulating hemocytes, neural ganglia, nerve fibers, neurosecretory, glial cells, gonads, stomach subcuticulum, heart, and antennal gland [111].

YHV-1 can cause lethal infections in farmed penaeid shrimp species, but some wild shrimp and crab species can be YHV-1 carriers and transmit the disease without showing serious symptoms themselves [102]. YHV-1 can be horizontally transferred when the YHV-1 virus is released into the water, or through a formula of the infected shrimp individual [95]. It has been reported that YHV-1 can remain infectious for at least 72 h in seawater, and that approximately 30 ppm of calcium hypochlorite is an effective disinfectant [103]. YHV-1 is combined with a specific receptor, YRP65 on the cell membrane of lymphocyte cells as its primary target organ [92]. Although there is no direct report that YHV-1 propagates vertically, it has been experimentally verified for GAV [1]. GAV was detected in infected mature ovarian and spermatophores in broodstock, fertilized eggs and nauplii from shrimp infected with GAV, which demonstrated efficient vertical propagation from both males and females [100].

Mortality in shrimp infected with YHV-1 occurs a few days after the onset of symptoms. Generally, individuals die within 1–2 days, and mass death (70–100%) occurs within 2–3 days [102,112]. YHV-1 infection can occur from the late post-larvae stage of development, but mass mortality usually occurs in the early to late juvenile stages [100]. In contrast, GAV causes death after 7–14 days in experimentally infected *P. monodon*, and mainly occurs as a chronic farm disease [95]. It was reported that there was 100% prevalence of GAV infection in healthy *P. monodon* in eastern Australia and common prevalence in healthy *P. monodon* in Vietnam and Thailand [108]. GAV-infections are much less lethal for shrimp than YHV-1, and mortality progresses more slowly, with100% mortality being rare. GAV-infected moribund shrimp do not show the pale discoloration typical of yellow head disease and are reddish [1]. Walker and Mohan (2009) [1] reported that YHV-1 was 106 times more virulent than GAV at lethal concentrations of 50% in an artificial YHV-1 and GAV challenge experiment.

There are various techniques for YHV detection, including reverse transcriptase-polymerase chain reaction (RT-PCR), nested RT-PCR (IQ2000™ YHV Detection and Prevention System), loop mediated isothermal amplification (RT-LAMP), in situ hybridization, and real time RT-LAMP, all of which are currently being used [113] (Table 5). PCR-based methods for detecting YHV-1 and GAV have high efficiency in terms of speed, sensitivity and specificity, and quantitative real-time RT-PCR using a TaqMan probe or SYBR Green chemistry are effective detection methods [114] (Table 5). The OIE manual recommends detection using the YHV ORF1b gene region to diagnose YHV [91]. YHV infection is histologically accompanied by the observation of pyknotic and karyorrhectic nuclei and dense basophilic cytoplasmic inclusions in the lymphoid organs and gills, as well as the target tissues such as hepatopancreas, hematopoietic tissue, heart, midgut, nerve cord, eyestalks, abdominal muscle, and soft head tissues [102,110] (Figure 17).

Prevention of YHV gene expression is considered a major method to control YHV infection; the method by RNA interference (RNAi)-based anti-YHV efficiency through dsRNA injection was reported to specifically inhibit YHV infection by inducing the sequence-specific degradation of mRNA [112]. Sanitt et al. (2014) [115] confirmed that three types of orally delivered dsRNA (dsRab7, dsYHV, combined dsRab7 + dsYHV) were effective in reducing mortality by YHV infection up to 70% compared to control (dsRab7: 70%, dsYHV: 40%, combined dsRab7 + dsYHV: 56%). YHV disease control should mainly be done through the selection of YHV-1 SPF individuals through PCR screening of broodstock and seeds, strengthening of biological security and sanitation measures in the farm, and management of the water environment [100].

### 3.3. Taura Syndrome Virus (TSV)

TSV (Taura syndrome virus) is known as one of the three most critical shrimp viruses alongside WSSV and YHV, as it has seriously damaged the shrimp aquaculture industry worldwide over the past two decades [95,116]. The name, TSV disease, comes from the Taura River in Ecuador, where it was first reported [52] in the *P. vannamei* of Ecuador in June 1992 (viral etiology confirmation in 1995). It has since spread to the Americas (Ecuador, Columbia, Honduras, USA, and Mexico), Asia (Thailand, Indonesia, China, Taiwan, and Myanmar), Africa, and the Middle East (Saudi Arabia), with new TSV strains continuing to appear as the virus adapts to new penaeid species and environments [117]. It is estimated that TSV in the Americas has resulted in 1.2 to 2 billion USD in economic losses from 1992–1996 [118].

TSV causes severe mortality in *P. vannamei* raised in the Americas. It is transmitted through regional and international migration of live host-larvae and broodstock [119]. TSV was originally limited to the Americas, but after *P. vannamei* was introduced to Asia, it was reported across Asia, in countries such as Thailand, Taiwan, and China and was spread via infected *P. vannamei* from Latin America [52]. TSV was first reported in juvenile *P. vannamei* in Ecuador in 1992 and then spread to Colombia in 1993, Honduras and Hawaii in 1994, Mexico and Guatemalan borders in 1995, Taiwan in 1998–1999, Thailand 2003, Korea and Texas coastal countries in 2004, Venezuela in 2005, Saudi Arabia in 2010–2011 and Venezuela in 2016 [1,120,121,122,123,124,125,126,127]. Since the first case of TSV infection in Asia was reported in *P. vannamei* imported for aquaculture from Taiwan in 1998, it has been reported in all Asian countries that import *P. vannamei* [62]. TSV was listed as an OIE-designated disease in 2000 and is widespread especially in the Americas and Asia [128] (Figure 18). TSV occurs in all regions except Australia, Africa and some specific regions according to the guidelines of the OIE Aquatic Animal Health Code, and it is the second most damaging disease in the shrimp aquaculture industry after WSSV, in terms of economic loss [2]. However, recently, through enhanced biological security measures, the introduction of TSV-SPF (specific pathogen free) species, and the production of TSV-resistant *P. vannamei*, the occurrence and damage caused by TSV infection has greatly been reduced [118].

TSV is a positive-sense, icosahedral-shaped, non-enveloped single-stranded RNA genome of 10.2 kb with a diameter of 32 nm [129] (Table 1). TSV is taxonomically classified in the Aparavirus genus belonging to the Dicistroviridae family [117]. TSV infects tissues of ectodermal and mesodermal origin, particularly hematopoietic tissue, epidermal epithelium, antennal glands, subcuticular connective tissue, lymphoid organs, and striated muscle [1]. The TSV viral capsid consists of three major polypeptides, VP1 (55 kDa), VP2 (40 kDa), and VP3 (24 kDa), and a minor polypeptide, VP0 (58 kDa) [130]. The TSV genome includes ORF 1 [the sequence motifs for non-structural proteins containing protease, helicase, and RNA-dependent RNA polymerase (RdRp); 6324 nt long, encoding a 2107 amino acid polyprotein with a 324 kDa molecular mass] and ORF 2 [the sequences for TSV structural proteins such as three major capsid proteins [VP1 (55 kDa), VP2 (40 kDa), and VP3 (24 kDa)]; 3036 nt long, encoding a 1011 amino acid polypeptide with a 112 kDa molecular mass [2] (Table 2). As the VP2 (40 kDa) gene among the capsid protein genes exhibits the highest genetic variation, it is widely used to determine the genetic relationship between TSV geographical isolates [117].

Phylogenetic analysis of TSV isolates has identified seven lineages, corresponding to geographic origins: (1) America such as Ecuador, Columbia, Honduras, USA, and Mexico from 1993–1998; (2) Southeast Asia (Thailand, Indonesia, China, Taiwan, snd Myanmar); (3) Mexico; (4) Belize; (5) Venezuela, (6) Colombia, and (7) Saudi Arabia [116] (Table 3). Based on the sequence of the VP1 (55 kDa) structural protein, three genotypic variants were identified: the American group, the Southeast Asian group, and the Belize group [52]. When the TSV isolate from Belize (GenBank no. AY826051-826053) in 2002 was compared with the reference isolate from Hawaiian (GenBank no. AY826054-826055), it was confirmed that the Belize isolate was a unique variant of TSV [117]. A new TSV genotype was observed in Saudi Arabia (GenBank no. JX094350), which was a distinct TSV isolate when compared to those from Southeast Asia and Latin America, and it shared 90% sequence identity with a reference isolate in Hawaii (GenBank no. AF277675) [122]. Phylogenetic analysis of Korean TSV strains based on the partial nucleotide sequence of VP1 (55 kDa) determined that Korean isolates (GenBank no. DQ099912-DQ099913) are closely associated with Thailand TSV types (GenBank no. AY912503-9125038) [131]. Sequence identity of TSV isolates for the Texas isolate (GQ502201) were very high in the Chinese and Thai isolates (GenBank no. DQ104696 and AY997025, respectively) and the Hawaii and Belize isolates (GenBank no. AF277675 and AY590471, respectively) (sequence identities for the Texas isolate ORF 1: 98% for the China and Thailand isolates, 97% for Hawaii and Belize isolates, sequence identities for the Texas isolate, an intergenic region (IGR) sequence: 98% for the Hawaii, China, Belize and Thailand isolates, sequence identities for the Texas isolate ORF 2: 97% for the Hawaii, China, and Thailand isolates, 96% for the Belize isolate) [132].

Other species susceptible to TSV infection include the Gulf white shrimp, *P. setiferus* and Pacific blue shrimp, *P. stylirostris*, which has been shown to be affected by TSV disease in the juvenile and adults, as well as in the nursery or post larval stages [52]. Although *P. vannamei* is known to be the main infective host for TSV, several other penaeid species (*P. stylirostris*, *P. setiferus*, *P. aztecus*, *P. duorarum*, *P. chinensis,* and *P. monodon*) have also been identified as susceptibility through experimental challenge infections. In addition, natural infections of TSV were found in various species including *P. stylirostris*, *P. monodon*, *P. japonicus*, *M. ensis* and the freshwater shrimp, *M. rosenbergii* [1]. Dhar and Allnutt (2008) [130] reported that the susceptibility of penaeid shrimp species to TSV differs from species to species, and *P. vannamei* and *P. schmitti* cultured in the Americas are highly susceptible, whereas other penaeid shrimp species in the Americas such as *P. stylirostris*, *P. setiferus*, *P. duorarum,* and *P. aztecus* reported less sensitivity to TSV infection. TSV usually causes serious disease as it infects *P. vannamei* in the late post larval to early juvenile stages, between 15–40 days, but it can also induce serious diseases in both sub-adult and adult *P. vannamei* [95].

TSV infection in *P. vannamei* is divided into three stages: acute (7 days after infection with an asymptomatic phase of 2–5 days), transition (lasting 5 days after the acute stage), and chronic (survivors after molting) stages, with a mortality rate of 60–90% [86,133]. Clinical symptoms of acute TSV infection in farmed *P. vannamei* are characterized by a reddish body color (especially on the tail; uropods, and appendages induced by chromatophore expansion) and irregular black (melanization) spots under the cuticle layer, in addition to anorexia, an erratic swimming behavior, lethargy, soft cuticles, anorexia, flaccid bodies and opaque musculature [95,129] (Figure 19). Shrimp acutely infected with TSV persist for 1–10 days after infection, and exhibit TSV-specific histological lesions, and mortality occurs during or immediately after molting [134,135]. According to Dhar and Allnutt (2008) [130], TSV infection begins within 24 h and death peaks between 7–10 days, and naturally or experimentally surviving individuals with acute infections develop grossly visible, multifocal, melanized lesions on the cephalothorax, tail, and appendages [95]. The main target organs following TSV infection are the cuticular epithelium of the gills, appendages, hindgut, foregut, and general body cuticle, and the lesion can spread to the underlying subcuticular connective tissue and striated muscle, and even the hematopoietic tissue, antennal gland, testes, and ovaries can become infected.

The transition stage of TSV infection is characterized by melanized multifocal lesions of the cephalothorax and tail with reduced mortality, lethargy, and anorexia [95]. Histological features of TSV infected shrimp at the transition stage show the initiation of spheroid developments within the lymphoid organ (LO), normal-appearing LO arterioles (tubules) that demonstrate a diffuse TSV probe positive signal by in situ hybridization (ISH), and infrequent scattered acute phase epithelial lesions [95] (Figure 20). The stage from transition infection to chronic infection begins with the shedding of the melanized exoskeleton and resumption of the molt cycle [136].

The TSV chronic infection stage (or ‘recovery stage’) appears from 6 days after TSV infection and lasts for a period of 8–12 months in experimentally infected *P. vannamei* with no disease symptoms, normal swimming behavior, and feeding, and no mortality [95]. During chronic TSV infection, there can be complete removal of TSV through apoptosis or there can be continued infection in a chronic state due to continuous virus replication, which is determined by the host’s immunity, nutritional status, and overall health condition [129]. In the chronic stage of TSV infection, shrimp are asymptomatic, and the only histologically identifiable lesions are numerous lymphoid organ spheroids (LOS) [133]. Surviving individuals after TSV infection can act as life-long carriers of TSV infection, and the prevalence of TSV infection in farms can vary from 0–100% [134].

TSV can maintain pathogenicity in dead shrimp for up to 3 weeks, and transmission of TSV can occur when healthy shrimp ingest infected moribund or dead *P. vannamei* through formula. The water-borne transmission of TSV has been experimentally shown to occur for up to 48 h after the period of maximum mortality, and it is known that TSV infection can be transmitted to other farms through the excrement of birds including seagulls, *Larus atricilla* that eat TSV-infected shrimp, as well as a flying aquatic insects such as water boatmen, *Trichocorixa reticulata* [52,130]. Transboundary transport of TSV occurs primarily through the sale and export of live post-larvae or adult shrimp infected with acute or chronic TSV, while frozen shrimp can also be potential carriers due to the ability of TSV to remain infective during prolonged freezing [95]. Although studies on the survival and resistance of TSV under environmental conditions are insufficient, it has commonly been shown to be very resistant, especially in seawater [52]. Although it is hypothesized that vertical transmission from TSV-infected broodstock to offspring is possible, it has not been experimentally verified [137].

*P. vannamei* infected with TSV exhibits a cumulative mortality rate of 60–95% (cumulative loss 80–95%, survival rate of ≥60%) within one week of TSV disease onset [52,95]. In the years following the first outbreak of TSV in Colombia, the mortality rate from TSV reached 100% [138]. According to Wertheim et al. (2009) [127], it was reported that mortality rates ranged from 40% to 100% when TSV infection occurred in *P. vannamei* farms. TSV infection occurs most frequently in *P. vannamei* in the nursery- the grow-out-stage post-larvae or in juveniles weighing <0.05–5 g within 14–40 days [62]. Efforts of several research and commercial breeding programs through TSV-SPR (specific pathogen resistance) selective breeding to control TSV disease since the mid-1990s have significantly reduced TSV incidence (Sookruksawong et al. 2013). Indeed, from 1999 to 2004, there were no TSV outbreaks in the shrimp farms of Colombia, indicating the success of a TSV-resistant breeding program in which 100% of the animals raised were TSV-SPR [138].

Diagnosis of pathogens following TSV disease infection is important to control, predict, and prevent potential outbreaks and significant economic losses [120]. TSV infection at acute, transition, and early chronic stages can be accurately diagnosed using histological or molecular methods, but it is difficult to detect low virus levels during the chronic stage, when the symptoms and most histological lesions disappear [86]. TSV virus testing is carried out using PCR assays, such as a commercial nested RT-PCR kits and reverse transcriptase PCR (RT-PCR) using TSV virus target organs such as uropods, gills, body cuticles, and swimming feet; the OIE recommends using a one-step PCR method for TSV testing [129,139] (Table 5).

In the acute stage of TSV, the cuticular epithelium of the appendages, gills, hindgut, foregut, and general body cuticle are infected as major target tissues, and infected cells appear to have highly basophilic pyknotic, karyorrhectic nuclei, and vivid cytoplasmic eosinophilia, with staining and sized cytoplasmic inclusion bodies in a variable manner [130]. The TSV at the transition stage histologically represents the onset of lymphoid organ (LO) arterioles (tubules) and spheroid development within the LO, and the marked histological characteristic during the chronic stage of infection is the LO spheroid appearance; spheroids include phagocytic semigranular and granular hemocytes undergoing apoptosis [130]. TSV control methods would be effective using farm-level biological security and TSV-specific pathogen free (SPF) and TSV-specific pathogen resistance (SPR) shrimp, a clean environment, and strict seed selection in addition to the immune system improvements for shrimp, could help to reduce the rate of TSV infection [123].

### 3.4. White Tail Disease (WTD)

WTD (white tail disease) is caused by *Macrobrachium rosenbergii* nodavirus (*Mr*NV) and extra small virus (XSV), and it induces critical economic losses, especially at the hatchery and nursery stages [140]. WTD was first reported in Guadeloupe (French West Indies) in 1995 or 1997 (named white tail disease from Pointe Noire, Guadeloupe in 1997) and later in Martinique (French West Indies) (1999), China (2003), India (2004), Thailand (2006), Taiwan (2006), Australia (2008), Malaysia (2012) [141,142,143,144] (Figure 21). White-tailed disease occurs in the freshwater shrimp *M. rosenbergii*, which is cultivated in many countries, and has an extremely high mortality rate (often reaching 100%) and causes enormous economic loss [145].

Natural infection of WTD was also observed in *P. monodon* and *P. indicus* hatcheries, which are geographically close to the freshwater shrimp *M. rosenbergii* hatcheries with reported WTD infections; the transmission of *Mr*NV and XSV from *M. rosenbergii* to *P. monodon* and *P. indicus* [144]. Mass mortality due to WTD occurs frequently in *M. rosenbergii* hatcheries in India, and the cumulative losses are estimated to be worth of millions of dollars [146]. WTD causes high mortality (up to 100%) in *M. rosenbergii* post-larvae within 2–3 days after infection. In India, WTD caused more than 50 freshwater shrimp hatcheries to have losses of 50%, which resulted in economic losses of approximately 15 million USD per year [147]. WTD (MrNV) causes large amounts of damage in all countries with aquaculture practices for *M. rosenbergii,* including the world’s largest producer, China [148]. This disease has the potential to disrupt the *M. rosenbergii* aquaculture industry in the future, and it was listed as the OIE-designated disease of 2009 [149].

WTD is caused by *Mr*NV (*Macrobrachium rosenbergii* nodavirus) which is accompanied by another virus, XSV (extra small virus) [142] (Table 1). *Mr*NV is a small icosahedral with non-enveloped two single-stranded RNA virus (RNA1: size 2.9 kb, RNA2: size 1.26 kb) approximately 26–27 nm in diameter and was observed in the cytoplasm of connective cells classified into the family Nodaviridae, which consists of two genera, Alphanodavirus and Betanodavirus, Nodaviruses have T = 3 capsids of a single polypeptide that is 43 kDa [54,144]. The phylogenetic tree obtained from RdRp demonstrates that *Mr*NV is more related to alphanodaviruses, whereas in the capsid-based phylogenetic tree, *Mr*NV and *Pv*NV (a second prawn nodavirus; *Penaeus vannamei* nodavirus) are more closely related to betanodaviruses (*Mr*NV and *Pv*NV: 69% homology in the capsid protein genes) [150,151]. Since it is difficult to classify *Mr*NV as an Alphanodavirus as it mainly infects insects and Betanodavirus which mainly infects fish, it has been proposed that it be classifies as a Gammanodavirus genus belonging to the Nodaviridae family [146,150]. Shrimp infected with *Mr*NV target hemocytes and myonuclei in the lower abdomen, they then spread to the rest of the abdomen, and subsequently, throughout the body via the hemolymph circulatory system, thereby observing the almost tissues of infected shrimp except for hepatopancreas and eyestalks [142]. *Mr*NV, a viral particle with an initial diameter of 27 nm, was observed in WTD-infected shrimp, and shortly thereafter, a second type of virus particle with an abnormally small diameter of 15 nm was observed in the WTD-infected shrimp tissue, which was named XSV [152]. Although there is evidence that *Mr*NV has a critical role in the pathogenesis of WTD, the role of XSV is also important in its pathogenesis [149]. XSV is an icosahedral and linear single stranded positive-sense RNA genome of 0.9 kb (approximately 700–1200 nucleotides) coding for a capsid protein, cp-17 with a 15 nm diameter that was identified in the cytoplasm of connective tissue cells [140]. *Mr*NV and XSV are found to be related in WTD-infected *M. rosenbergii*, but the interactions between the two pathogens and their effects on pathogenicity are currently unknown [149,150].

*Mr*NV genomic nucleotide sequencing suggested that RNA-1 contained 3202 nucleotides (GenBank no. AY222839) and RNA-2 consisted of 1175 nucleotides (GenBank no. AY222840) [153] (Table 2). RNA-1 included two nonstructural proteins such as A protein [RNA-dependent RNA polymerase (RdRp) containing approximately 1000 amino acids (ca. 100 kDa)] and B protein [13 kDa encoding 30 region of RNA-1 (2725–3126 nucleotides)], whereas RNA-2 included a single polypeptide in the capsid protein [54]. XSV genomic nucleotide sequencing indicated that it consisted of 796 nucleotides such as the coding sequence of the capsid protein CP-17 (17 kDa) and CP-16 (16 kDa) [137]. The *Mr*NV structural protein consisted of a single protein of approximately CP-43 (43 kDa), whereas two polypeptides of approximately CP-17 (17 kDa) and CP-16 (16 kDa) were observed in the XSV particles [150].

Phylogenetic analysis of the WTD isolates was divided into groups for the French West Indies, China, India, Taiwan, Malaysia, Australia, Thailand, and France. The complete genome sequence of *Mr*NV RNA-1 and RNA-2 was reported in 2003 (French West Indies, Gen bank no. AY222839 and AY222840, respectively) in 2004 (Australia, GenBank no. JN619369 and JN619370) [143,154]. Analysis of the nucleotide sequence was used to determine identity with other *Mr*NV. The nucleotide sequence of *Mr*NV (RNA-1) isolated India (GenBank no. AAO60068) has 98% identity with *Mr*NV isolated from French West Indies (GenBank no. AY222839). Similar to *Mr*NV, the nucleotide sequence of XSV isolated from Taiwan (GenBank no. DQ521573) has 97% and 98% identity with the XSV isolated from India (GenBank no. AY247793) and China (GenBank no DQ147318), respectively [151]. In addition, that isolated from Australia (Australian, GenBank no. JN619369) has 94%, 95%, 95%, and 97% identity with *Mr*NV isolated French West Indies (GenBank no. AY222839), China (Chinese 1, GenBank no. AY231436; Chinese 2, GenBank no. FJ751226) and Malaysia (GenBank no. JN187416), respectively. The nucleotide sequence of *Mr*NV (RNA-2) isolated from Australia (GenBank no. JN619370) has 92% identity with French West Indies (GenBank no. AY222840), Chinese 2 (GenBank no. FJ751225), China (GenBank no. AY231437), and Thailand (GenBank no. EU150126-150129) [143].

*M. rosenbergii* is more susceptible to WTD than other shrimp species, and especially in the larvae, post-larvae, and juvenile stages of development, it has a high mortality. In post-larvae infected *M. rosenbergii*, the striated muscles of the cephalothorax, abdomen and tail are the most targeted tissues, and adults of *M. rosenbergii* infected with WTD are resistant to WTD and function only as carriers [140]. Although *M. rosenbergii* was initially reported as the only host species for the onset of WTD induced by *Mr*NV and XSV, subsequent reports confirmed that marine shrimp species such as *P. indicus*, *P. japonicus*, *P. monodon,* and *P. vannamei* at the post-larval (PL) stage are also susceptible and capable of high mortality [150] (Table 4). However, according to Bonami and Widada (2011) [150], in the WTD challenge test by the oral route and injection, marine shrimp such as *P. indicus*, *P. japonicus,* and *P. monodon* did not show high susceptibility to the WTD and had no clinical signs or mortality.

Clinical signs of WTD-infected shrimp include lethargy, degeneration of the telson and uropods, opaqueness of the abdominal muscle, reaching up to 100% within 4 days of onset [150,155] (Figure 22). WTD-infected shrimp at post-larvae stage develop symptoms in the second or third abdominal region, gradually extending from the center of the muscle to the anterior and posterior parts of the muscle, showing lethargy and opaqueness of the abdominal muscle [156]. WTD infection begins in some areas of the tail, extends to the tail muscles (abdomen), and causes whitish pigmentation in all muscles in the final stage, including the head (cephalothorax) muscles; in severe cases, degeneration of telsons and uropods is observed [147,150]. WTD symptoms mainly appeared when *Mr*NV values were high, suggesting that *Mr*NV plays an important role in WTD [140].

*Mr*NV and XSV can be transmitted horizontally in the form of dead tissue, live carriers, and free virions through formulas of *M. rosenbergii* infected with WTD, and natural hosts of adjacent ecosystems and culture systems [142,146]. In the WTD horizontal transmission experiment, artemia was exposed to *Mr*NV and XSV by immersion and oral routes, confirming that it could act as a reservoir or carrier for the *Mr*NV and XSV [140]. A high prevalence of WTD induced by *Mr*NV and XSV has been reported in hatchery larvae and post-larvae of *M. rosenbergii*, suggesting that vertical transmission may occur from infected brooders to offspring during spawning [157]. Murwantoko et al. (2016) [147] also reported the vertical transmission of *Mr*NV and XSV in *M. rosenbergii*, suggesting that this is the main disease transmission mechanism of WTD. Vectors of WTD include penaeid shrimp (*P. japonicus*, *P. indicus*, and *P. monodon*), aquatic insects (*Cybister* sp., *Aesohna* sp., *Belostoma* sp., and *Notonecta* sp.), and artemia [158]. A WTD challenge experiment using both oral and intramuscular routes in *M. malcolmsonii* and *M. rude* did not cause clinical symptoms or mortality but indicated that it could serve as a reservoir as the toxicity of *Mr*NV and XSV were maintained [147].

Mortality due to WTD infection reaches its maximum 5–6 days after the first severe symptoms appear, and infected post-larvae die within 15 days, and surviving post-larvae can grow to market size just like normal individuals [140]. *Mr*NV infection of *M. rosenbergii* at the post-larvae stage results in a high mortality rate of almost 100% but it is not fatal for adults [139]. Bonami and Widada (2011) [150] reported that mortality started to occur 1–3 days after the first clinical signs of post-larvae *M. rosenbergii* infection with WTD, and the cumulative mortality rate reached 100%, 8–14 days post-infection.

To confirm WTD infection, real-time RT-PCR is one of the most sensitive diagnostic methods and has been used to detect the presence of both *Mr*NV and XSV [149] (Table 5). Of the many samples infected with WTD, the majority of *Mr*NV-positive samples were also positive for XSV, but some samples did not have XSV, and in some cases XSV was detected without *Mr*NV [150]. Histological features of WTD-infected shrimp include large oval or irregular basophilic cytoplasmic inclusions with a diameter of 1–4 µm in the infected muscles of the abdomen, cephalothorax, and intratubular connective tissue of the hepatopancreas [140]. Murwantoko et al. (2016) [147] also found lesions in the muscle and connective tissues upon histological examination of the shrimp infected with WTD, and these lesions corresponded to the dense basophilic inclusions that had a diameter of 0.5–3.0 µm, and were located in the cytoplasm. Jariyapong et al. (2018) [159] confirmed coagulation necrosis of skeletal muscle in *P. vannamei* infected with *Mr*NV (Figure 23B). Hayakijkosol et al. (2011) [160] reported that muscle degeneration, tissue necrosis, and myolysis with hemocytic infiltration were found in MrNV-infected redclaw crayfish, *Cherax quadricarinatus* (Figure 23C,D).

To control the spread of WTD, it is essential to develop highly sensitive and rapid diagnostic methods that can detect pathogens early, because effective methods such as vaccines or treatment for controlling and preventing WTD have not been presented [142]. Screening using sensitive diagnostic methods such as reverse-transcription polymerase chain reaction (RT-PCR) and enzyme-linked immunosorbent assay (ELISA) to select specific pathogen free (SPF) brood stock and post-larvae can be an effective method [153]. Since virus-borne infections such as WTD are difficult to control, only preventive measures, including daily monitoring of shrimp health and early diagnosis, are critical and can help manage the WTD occurrence [149].

**Table 1 viruses-14-00585-t001:** Summary information for DNA and RNA viral diseases infections.

Virus Type		Pathogen	Taxonomy		Morphology	Reference
DNA virus	ds DNA	WSSV	Family	*Nimaviridae*	● Rod-shape to elliptical ● Tail like appendage at one end of the virion ● Virion size: 80–120 × 250–380 nm ● Envelope: - Tadpole-shaped spike - Thickness: 6–7 nm ● Nucleocapsid: - 15 helices composed of 14 globular capsomers along its long axis - Ring structure at one terminus - Size: 54–85 × 180–440 nm	[9,13,16,18,20,22,52]
			Genus	*Whispovirus*		
	ss DNA	IHHNV (*Decapod penstylhamaparvovirus* 1)	Family	*Parvoviridae*	● Virus diameter: 20–22 nm ● Containing a 4 kb linear ssDNA genome ● Density: 1.40 g/mL in CsCl ● Non-enveloped ● Icosahedral shape ● Smallest penaeid shrimp virus ● Density: 1.40 g/mL in CsCl ● Capsid - Four polypeptides with molecular masses of 74 K, 47 K, 39 K, and 37.5 K, respectively	[57,59,60,65,66,85,161,162]
			subfamily	*Hamaparvovirinae*		
			Genus	*Penstylhamaparvovirus*		
RNA virus	ds RNA	IMNV (PsIMNV)	Family	*Totiviridae*	● Virus diameter: 40 nm ● Virion size: 83,226–83,230 bp ● Density: 1.366 g/mL in CsCl ● Non-enveloped ● Icosahedral shape ● Tridimensional image reconstruction of the IMNV virion revealed a 120 kDa capsid protein that has a totivirus-like architecture ● Genome consists of a double-stranded RNA molecule that is 7561–8230 bp in size	[73,77,79,83,85]
			Genus	Similar *Giardiavirus*		
	ss RNA	YHV	Order	*Nidovirales*	● Rod-shape ● Envelope: - Contain two transmembrane glycoprotein (gp64 and gp 116) ● Size: 40–60 nm × 150–200 nm ● Buoyant density in sucrose: 1.18–1.20 g/mL ● Nucleocapsid - Helical symmetry - Composed of a coiled filament - Diameter: 16–30 nm - Periodicity: 5–7 nm	[68,95,104]
			Family	*Roniviridae*		
			Genus	*Okavirus*		
		TSV	Order	*Picornavirales*	● Icosahedral shape ● Non envelope ● Virion diameter: 30–32 nm ● Buoyant density: 1.337–1.338 g/mL	[2,62,95,116,117]
			Family	*Dicistroviridae*		
			Genus	*Aparavirus*		
		WTD (*Mr*NV)	Famliy	*Nodaviridae*	● Virus diameter: 26–27 nm ● Density: 1.27–1.28 g/mL in CsCl ● Non-enveloped ● Icosahedron shape ● Consists of two pieces: RNA1 and RNA2 ● Capsid contains a single polypeptide of 43 kDa ● Located in the cytoplasm of infected target cells, particularly connective tissue cells	[56,140,141,142,144,145,146,149,150,157,163]
			Genus	*Gammanodavirus*		
		WTD (XSV)	Unassigned		● Virus diameter: 14–16 nm ● Non-enveloped ● Icosahedral shape ● Located in the cytoplasm of infected target cells, particularly connective tissue cells	

**Table 2 viruses-14-00585-t002:** Summary of the DNA and RNA viral diseases ORF characteristics.

Virus Type		Pathogen	ORF	Characteristics	Reference
DNA virus	ds DNA	WSSV	ORF75	● Number of bp in the repeat unit: 45 bp (type 1), 102 bp (type 2) ● Repeat unit sequences: - Type 1: GAA GCA GCT CCC CCA CTT AAA GGT GCA CTT GGA CGT AAG AGG CGC - Type 2: GAA GCA GCT CCC CCA CTT AAA GGT GCG CTT GGA CGT AAG AGG CGC GAA GCA GAA TCC TTG GAG GAA GAA CTT GTG TCT GCT GAA GAA GAA CGT GAA AAG CGC ● Primers: - ORF75F (5′-GCC AGA TTT CTT CCC CTA CC-3′) - ORF75R (5′-CTC CAT GTA GAG GCA AAG CA-3′)	[9,13,20,24,164,165,166,167,168,169,170,171,172]
			ORF94	● Number of bp in the repeat unit: 54 bp ● The most informative single genetic marker ● Repeat unit sequences: - CGC AAA AAG CGT GCC GCA CCT CCA CCT GAG GAT GAA GAA GAG GAT GA G/T TTC TAC ● Primers: - ORF94-F (5′-TCT ACT CGA GGA GGT GAC GAC-3′) - ORF94-R (5′-AGC AGG TGT GTA CAC ATT TCA TG-3′)	
			ORF125	● Number of repeat unit: 69 bp ● Repeat unit sequences: - AG/TA AAC AAG GAG GAA GAA GAC GCG AGG ATA AAG CGT GTA GCC GTC AGG ACA TTT ACA GCC ATC AGA GAAA ● Primers: - ORF125F (5′-TGG AAA CAG AGT GAG GGT CA-3′) - ORF125R (5′-CAT GTC GAC TAT ACG TTG AAT CC-3′)	
			ORF14/15	● Prone to the recombination region	
			ORF23/24	● Deletion region	
			ORF109	● Nucleotide position: 163996–164238 ● VP15 - Location: nucleocapsid - Overlaps with ORF110 (11 kDa)	
			ORF182	● Nucleotide position: 290363–289998 ● VP19 - Location: envelope	
			ORF153	● VP26 - Location: nucleocapsid - Tegument protein - N-terminal anchors in the envelop - C-terminal is bound to the nucleocapsid - Capable of binding to actin or actin-associated proteins - Interacts with VP51	
			ORF-wsv002	● VP24 - Location: nucleocapsid - Major structural protein - Chitin-binding protein	
			ORF-wsv421	● VP28 - Location: envelope - Major structural protein - Early stages of virus infection - Viral attachment protein - Helps the virus to enter the cytoplasm	
			ORF-wsv308	● VP51 - Location: nucleocapsid - Molecular mass: 51.9 kDa - Encodes a 466 aa protein	
	ss DNA	IHHNV (*Decapod penstylhamaparvovirus* 1)	ORF1	● Length: 2001 bp ● Starts at nt 648 and terminates with a TAA codon at 2648 nt ● Encodes a 666 aa protein with a molecular weight of 75.77 kDa ● Coding domain - Nonstructural proteins 1 - Function: enzymatic activities involved in viral transcription and replication ● Contained highly conserved replication initiator motifs (rolling-circle replication (RCR) motifs) and NTP-binding and helicase domains (ATPase motifs)	[57,58,59,85,173,174]
			ORF2	● Length: 1092 bp ● Starts with an ATG codon at 591 nt and terminates with a TAG codon at 1681 nt ● Encodes a 363 aa protein with a molecular mass of 42.11 kDa ● Coding domain - Nonstructural proteins 2 - Function: viral multiplication	
			ORF3	● Length: 990 bp ● Smallest among the three ORFs ● Starts with an ATG at 2590 nt, and terminates with an TAA codon at 3577 nt ● Encodes a 329 aa protein with a molecular mass of 37.48 kDa ● Coding domain: CP	
RNA virus	ds RNA	IMNV (PsIMNV)	ORF1 (59 ORF)	● Length: 5127 nt ● Nucleotide: 136–4953 ● First half of ORF1 - Region of the RNA-binding protein - Contained a dsRNA-binding motif in the first 60 aa ● The second half of ORF1 - Encodes a capsid protein (molecular mass of 106 kDa)	[62,73,74,76,83,86]
			ORF2 (39 ORF)	● Length: 2739 nt ● Nucleotides: 5241–7451 ● Encoded a putative RNA-dependent RNA polymerase (RdRp)	
	ss RNA	YHV	ORF1a	● Nucleotides: 12,216 ● Encodes a 4027 aa polyprotein (pp1a) ● ORF1a polyprotein (pp1a) - 15 amino acids longer than GAV pp1a - Contains four hydrophobic domains (HD1, HD2, HD3 and HD4) - 3C-like cysteine protease catalytic domain - Papain-like protease (PL1) domain - Lacks the canonical α + β fold of the papain-like protease (PLX) domain - Autolytic activity	[1,92,100,104,107,111,114,163,175,176]
			ORF1a/ORF1b	● Fold into a complex pseudoknot structure ● A slippery hepta nucleotide (AAAUUUU) ● The ribosomal frame-shift (RFS): - Generate polyprotein pp1ab ● ORF1a/ORF1b overlaping polyprotein (pp1ab): - Overlaps by 37nt - 15 amino acids longer than GAV pp1ab	
			ORF1b	● Nucleotides: 7887 ● Encodes a 6688 aa polyprotein (pp1b) ● Encodes enzyme of the replication complex: - RNA dependent RNA polymerase - Cysteine and histidine-rich domain (C/H) Zn fingers - Helicase (HEL) - Exonuclease - Uridylate-specific endoribonuclease - Ribose-2′-O-methyl transferase domains ● Untranslated region (UTR) between ORF1b and ORF2: - 352 nt	
			ORF2	● Encodes a 146 aa nucleoprotein (p20)	
			ORF3	● Encodes a 1666 aa polyglycoprotein (pp3): - Generates the envelop glycoproteins 22, 64, and 116 ● Glycoprotein 22 (gp 22) - Unknown function ● Glycoprotein 64 (gp 64) - Major structural protein - Form the spike-like projection on the virion surface ● Glycoprotein 116 (gp 116) - Major structural protein - Form the spike-like projection on the virion surface - Bind to a 65 kDa protein in the lymphoid organ cells - Identity of gp116 with GAV gp116: 73% ● Untranslated region (UTR) between ORF3 and the 3‘-poly(A) tail - 677 nt	
			ORF4	● 677 nt region downstream of ORF3 ● Encodes a 20 aa polypeptide ● Interrupted by multiple stop codons	
		GAV	ORF1a	● ORF1a polyprotein (pp1a) - Encodes 3C-like cysteine protease catalytic domain - Identity of pp1a with YHV pp1a: 82.4%	[1,92,100,107,114,177]
			ORF1b	● Identity of the ORF1b sequence with YHV ORF1b: ~82% ● Untranslated region (UTR) between ORF1b and ORF2 - 93 nt	
			ORF1a/ORF1b	● Identity of pp1ab with YHV pp1ab: 84.9%	
			ORF2	● Encodes a 144 aa polypeptide	
			ORF3	● Encodes a 1640 aa glycoprotein ● Identity of the ORF3 sequence with YHV ORF3: ~74%	
			ORF4	● 638 nt region downstream of ORF3 ● An unidentified 83 aa polypeptide	
		TSV	ORF1	● Nucleotides: 6324 ● Amino acid polyprotein: 2107 ● Molecular mass: 234 kDa ● Encode the non-structural proteins - Helicase - Protease - RNA dependent RNA polymerase	[95,128,130,178]
			ORF2	● Nucleotides: 3036 ● Amino acid polyproteins: 1011 ● Molecular mass: 112 kDa ● Encodes 3 major and 1 minor capsid proteins: - Major VP1 (55 kDa) - Major VP2 (40 kDa) - Major VP3 (24 kDa) - Minor VP0 (58 kDa)	
		WTD (*Mr*NV)	ORF1 (RNA-1)	● Length: 2.9 Kbp ● Nucleotides: 3202 ● Encodes approximately 1000 amino-acids (approximately 100 kDa) and a B protein encoded by the 30 region (13 kDa) ● Coding domain: - Protein A or RNA-dependent RNA polymerase - Protein B2	[56,140,150,153,156,157]
			ORF2 (RNA-2)	● Length: 1.26 Kbp ● Nucleotides: 1175 ● Coding domain: - Capsid protein (CP-43)	
		WTD (XSV)	XSV genome	● Length: 900 bp ● Nucleotides: 796 ● Short poly (A) tail and polyadenylation signal AAUAAA were found ● Coding domains: - Capsid protein (CP-16 or CP-17) - Methionine N-terminal ends for both polypeptides	

**Table 3 viruses-14-00585-t003:** Summary isolation and GenBank accession number information for the DNA and RNA viral disease infections.

Type		Pathogen	Origin	Host Species	Isolation	ORF Region	GenBank No.	Year	Reference
DNA virus	ds DNA	WSSV	Mexico	*Penaeus vannamei*	Mx-F	Hypothetical protein (ORF13 and ORF16) gene; Nonfunctional hypothetical protein gene	HQ257380	2001	[179]
					Mx-H		HQ257381	2004	
					Mx-C	Nonfunctional hypothetical protein genes	HQ257382	2005	
					Mx-G		HQ257383	2004	
					Mx-L1		HQ257384	2001	
					WSSV-MX08	Complete genome	KU216744	2008	[33]
				*Penaeus vannamei*	LG	Partial genome	MG432482	2012	[180]
					JP		MG432479	2011	
					AC1		MG432474	2011	
					DV1		MG432477	2011	
					LC1		MG432481	2011	
					LC10		MG432480	2011	
					ACF2		MG432475	2012	
					ACF4		MG432476	2012	
					GVE05		MG432478	2005	
			India	*Penaeus monodon*	ANI	wsv285 gene	KX980155	2016	[181]
					WSSV-IN-07-I	Unknown gene	EF468499	2007	[182]
					WSSV-IN-06-I		EF468498	2006	
					WSSV-IN-05-I		EU327499	2005	
					WSSV-IN-05-II	ORF23/ORF24 region genomic sequence	EU327500	2005	
				*Penaeus vannamei*	IN_AP4RU	Complete genome	MG702567	2013	[38]
			Iran	*Penaeus vannamei*	IRWSSVKH2	Hypothetical protein 75 gene	KF157839	2012	[183]
					IRWSSVKH4	ORF75 gene	KC906268	2011	
					IRWSSVKH5		KF157833	2012	
					IRWSSVKH3		KF157832	2012	
					IRWSSVSIS3		KP455493	2014	
					IRWSSVSIS2		KF956791	2013	
				*Penaeus indicus;* *Penaeus vannamei*	IWV-MS21	ORF75 gene	KX694234	2013	
					IWV-MS24		KX694236	2014	
					IWV-MS25		KX694237	2014	
					IWV-MS26		KX694238	2014	
					IWV-MS19		KX694242	2013	
					IWV-MS18		KX584741	2013	
			China	*Penaeus japonicus*	WSSV-CN	Complete genome	AF332093	1996	[30]
					WSSV-CN01		KT995472	1994	[34]
				*Procambarus clarkii*	WSSV-CN02	Complete genome	KT995470	2010	[34]
					WSSV-CN-Pc		KX686117	2015	[36]
				*Penaeus vannamei*	WSSV-CN03	Complete genome	KT995471	2010	[34]
				*Marsupenaeus japonicus*	WSSV-CN04	Complete genome	KY827813	2012	[35]
			Thailand	*Penaeus monodon*	WSSV-TH	Complete genome	AF369029	1996	[29]
					TH-96-II	Nonfunctional ORF14 gene; ORFI, ORFII, ORFIII, ORFIV, and ORFV genes; ORF15 and ORF16 gene	AY753327	2005	[184]
			Taiwan	*Penaeus monodon*	WSSV-TW	Complete genome	AF440570	1994	[31]
			South Korea	*Penaeus vannamei*	WSSV-KR	Complete genome	JX515788	2011	[32,34]
			Australia	*Penaeus monodon*	WSSV-AU	Complete genome	MF768985	2016	[37]
			USA	*Penaeus vannamei*	CN_95_DFPE	Complete genome	MN840357	2017	[41]
			Ecuador	*Penaeus vannamei*	WSSV-EC-15098	Complete genome	MH090824	2015	[39]
			Brazil	*Penaeus vannamei*	WSSV-chimera	Complete genome	MG264599	2015	[40]
					FSL39	Partial genome	MF784752		
	ss DNA	IHHNV (Type I)	Australia	*Penaeus monodon*	Australian	Non-structural protein gene Non-structural protein 1 gene Capsid protein genes	GQ475529	2008	[60]
		IHHNV (Type II)	Thailand	*Penaeus monodon*	-	Non-structural protein 2 gene Non-structural protein 1 gene Capsid protein genes	AY362547	2003	[173]
					IHHNV_TH		AY102034	2000	[185]
			Taiwan	*Penaeus monodon*	Taiwan B	Non-structural protein 2 gene Non-structural protein 1 gene; Capsid protein genes	AY355307	2003	[186]
			Vietnam	*Penaeus monodon*	IHHNV-VN	Non-structural protein 2 gene Non-structural protein 1 gene Capsid protein genes	JN616415	2009	[60]
					ST		KC513422	2011	
				*Penaeus monodon;* *Penaeus vannamei*	KK-Lv-VIET1	Non-structural protein 1 gene	MN481525	2019	[187]
				*Penaeus stylirostris*	VN2007	Complete genome	KF031144	2007	[57]
			India	*Penaeus monodon*	IN-07	Complete genome	GQ411199	2007	[60]
					IHHNV	Capsid protein gene	FJ169961	2007	[173]
		IHHNV (Type III)	Vietnam	*Penaeus monodon*	KG	Complete genome	JX840067	2012	[57]
			Taiwan	*Penaeus monodon*	Taiwan A	Non-structural protein 2 gene; Non-structural protein 1 gene; Capsid protein genes	AY355306	2003	[186]
					Taiwan C		AY355308	2003	
			Ecuador	*Penaeus vannamei*	IHHNV	Non-structural protein 2 gene; Non-structural protein 1 gene; Capsid protein genes	AY362548	2003	[186]
			Brazil	*Penaeus vannamei*	IHHNV_BR	Partial genome	KJ862253	2013	[60]
			China	*Penaeus penicillatus*	IHHNV	Complete genome	KJ830753	-	[60]
				*Penaeus monodon*	Fujian		EF633688	2007	[188]
					Ganyu		JX258653	2009	[57]
				*Penaeus vannamei*	CSH-1		KF907320	2012	
				*Penaeus vannamei*	Sheyang		KF214742	2011	
			Hawaii	*Penaeus stylirostris*	Hawaii A	Complete genome	NC_002190	1990	[60]
					Hawaii B		AF218266	1990	
			Malaysia	*Macrobrachium rosenbergii*	IHHNV	Non-structural protein genome	HM536212	2009	[189]
			Taiwan	*Macrobrachium rosenbergii*	AC-04-367	Non-structural protein 1 gene	DQ057982	-	
					AC-05-005		DQ057983	-	
			Mexico	*Penaeus stylirostris*	IHHNV	Non-structural protein 2 gene; Non-structural protein 1 gene; Capsid protein genes	AF273215	2000	[190]
			South Korea	*Penaeus vannamei*	K1	Structural protein gene	HQ699073	2010	[191]
					K2		HQ699074	2010	
					KLV-2010-01	Complete genome	JN377975	2010	[58]
		IHHNV(Type A)	Madagascar	*Penaeus monodon*	IHHNV	Non-structural protein 1 gene; Structural protein genes; Unnamed retrotransposon reverse transcriptase gene	DQ228358	-	[191]
			Australia	*Penaeus monodon*	Au2005	Non-structural protein 2 gene; Non-structural protein 1-like gene; Viral capsid protein gene	EU675312	-	[188]
		IHHNV (Type B)	Tanzania Mozambique	*Penaeus monodon*	East Africa	Non-structural protein 1 gene; Structural protein genes	AY124937	2000	[185]
RNA virus	ds RNA	IMNV (PsIMNV)	Indonesia	*Penaeus vannamei*	ID-EJ-12-1	ORF1/ORF2 and ORF1 polyprotein genes	KJ636783	2012	[40,77]
					ID-EJ-12-1	ORF1 polyprotein	AIC34743	2012	
					ID-EJ-12-2	ORF1/ORF2	AIC34746	2012	
					ID-EJ-12-3	ORF1 polyprotein	AIC34749	2012	
					ID-LP-12-2		AIC34750	2012	
					ID-BB-12		AIC34752	2012	
					ID-EJ-06	Structural protein	ABN05324	-	
					ID-LP-11	Complete genome	KJ636782	2011	
					ID-LP-11	ORF1 polyprotein	AIC34741	2011	
					ID-LP-12-1	ORF1/ORF2	AIC34748	2012	
					IMNV	Complete genome	EF061744	-	[74]
					Indonesia		KF836757	2013	[192]
			Brazil	*Penaeus vannamei*	BZ-03	Structural protein	AAT67230	-	[77]
					ZS2011001	Capsid protein	AGF33812	2004	
					Brazil 01	Structural protein	ADG37656	2007	
					Brazil 02		ADN43996	2007	
					IMNV-BZ-11-UAZ219	ORF1 polyprotein	AIC34754	2011	
					IMNV	Complete genome	AY570982	-	[74]
	ss RNA	YHV (genotype 1)	Thailand	*Penaeus monodon*	YHV1992	Complete genome	FJ848673	1992	[98,101]
					YHV1995	Complete genome	FJ848674	1995	
					Chachoengsao 1998	Complete genome	EU487200	1988	[98,108]
					YHA-98-Ref	pp1ab gene	EU785033	1998	[98,114]
					Thailand: Cholburi	Envelope structural glycoprotein gene	EF156405	1999	[108]
					YHV1999	Complete genome	FJ848675		[98,101]
					YHV-PmA	3C-like protease gene	EU977577	-	[108]
						Replicase polyprotein 1ab gene	EU977578		
						RNA polymerase gene	EU977579		
						Helicase gene	EU977580		
						Nucleocapsid gene	EU977581		
						Glycoprotein 116 gene	EU977582		
						Glycoprotein 64 gene	EU977583		
						Genomic sequence	EU977584		
					THA-00-DRH	pp1ab gene	EU785032	2000	[98,114]
					THA-01-D4		EU785004	2001	
					THA-01-D8		EU785034	2001	
					THA-01-D9		EU785019	2001	
					THA-01-D10		EU784984	2001	
					THA-02-D34		EU785001	2002	
					THA-03-D1		EU784982	2003	
					THA-03-D2		EU784991	2003	
					THA-03-D3		EU784998	2003	
					THA-03-DB1		EU785023	2003	
					THA-03-D29		EU785035	2003	
					THA-03-D30		EU784999	2003	
					THA-03-D33		EU785000	2003	
				*Penaeus vannamei*	YHV	ORF1b genes	FJ627274	2007	[106]
			Mexico	*Penaeus vannamei*	YHV	3C-like protease gene	DQ978355	2000	[108]
						ORF1a and ORF1b polyprotein gene	DQ978356		
						Nonfunctional ORF1b polyprotein gene	DQ978357		
						ORF1b polyprotein gene	DQ978358		
						Helicase gene	DQ978359		
						Nucleocapsid gene	DQ978360		
						Glycoprotein 116 gene	DQ978361		
						Glycoprotein 64 gene	DQ978362		
						ORF4-like gene	DQ978363		
			China	*Fenneropenaeus chinensis*	Hb2012	Replicase polyprotein 1b mRNA	KF278563	2012	[98]
		GAV (genotype 2)	Australia	*Penaeus monodon*	GAV	Complete genome	AF227196	-	[98,101,108]
							NC_010306	-	[101]
					AUS-97-MCMS1	pp1ab gene	EU784980	1997	[98,114]
					AUS-97-MCMS2		EU784989	1997	
					AUS-97-MCMS3		EU785038	1997	
					AUS-00-H2		EU785029	2000	
					AUS-00-HL4		EU785030	2000	
					AUS-00-HL5		EU785031	2000	
					AUS-00-HL11		EU785028	2000	
					AUS-96-Ref		EU785026	1996	
			Vietnam	*Penaeus monodon*	VNT-01-H65	pp1ab gene	EU785039	2001	[114]
					VNT-01-H77		EU785013	2001	
					VNM-02-H6		EU785009	2002	
					VNM-02-H64		EU785008	2002	
			Thailand	*Penaeus monodon*	THA-03-HB3	pp1ab gene	EU785024	2003	[114]
					THA-03-HG		EU785025	2003	
					THA-03-HA		EU785021	2003	
					THA-03-HN		EU785022	2003	
					THA-04-H20		EU784992	2004	
					THA-04-HK		EU785027	2004	
		YHV (genotype 3)	Vietnam	*Penaeus monodon*	VNM-02-H5	pp1ab gene	EU785006	2002	[98,114]
					VNM-02-H258		EU784994	2002	
					VNM-02-H81		EU785016	2002	
					VNM-02-H70		EU785012	2002	
					VNM-01-H41		EU785040	2001	
					VNM-01-H42		EU785041	2001	
					VNM-02-H278		EU784996	2002	
					VNM-02-H264		EU784995	2002	
					VNM-02-H93		EU785020	2002	
					VNM-02-H93	p20 gene; pp3 gene	EU785042	2002	[114]
			Indonesia	*Penaeus monodon*	IDN-04-H7	pp1ab gene	EU785011	2004	[114]
					IDN-04-H11		EU784985	2004	
					IDN-04-H10		EU784983	2004	
					IDN-04-H4		EU785002	2004	[98,114]
			Malaysia	*Penaeus monodon*	MYS-03-H1	pp1ab gene	EU784981	2003	[114]
					MYS-03-H2		EU784990	2003	
					MYS-03-H3		EU784997	2003	
			Mozambique	*Penaeus monodon*	MOZ-04-H1	pp1ab gene	EU784986	2004	
		YHV (genotype 4)	Thailand	*Penaeus monodon*	YHV type 4	ORF1b polyprotein gene	EU170438	-	[98,193]
						gp116 gene	EU123854		
			Indonesia	*Penaeus monodon*	IND-02-H9	pp1ab gene	EU785017	2002	[98,114]
					IND-02-H5		EU785005	2002	
					IND-02-H7		EU785010	2002	
			India	*Penaeus monodon*	IND-02-H9	p20 gene; pp3 gene	EU785043	2002	[114]
		YHV (genotype 5)	Thailand	*Penaeus monodon*	THA-03-SG21	pp1ab gene	EU784993	2003	
					YHV	ORF1b polyprotein gene	EU853170	2005	[193]
			Malaysia	*Penaeus monodon*	MYS-03-H4	pp1ab gene	EU785003	2003	[114]
			Philippines	*Penaeus monodon*	PHL-03-H8		EU785015	2003	
		YHV (genotype 6)	Mozambique	*Penaeus monodon*	MOZ-04-H6	pp1ab gene	EU785007	2004	
					MOZ-04-H8		EU785014	2004	
					MOZ-04-H9		EU785018	2004	
					MOZ-04-H11		EU785036	2004	
					MOZ-04-H12		EU785037	2004	
		YHV (genotype 7)	Australia	*Penaeus monodon*	YHV7 (13-00169-01) PCR1	ORF1b polyprotein gene	KP738160	2012	[98,105]
					YHV7 (13-00169-01) PCR2		KP738161		
					YHV7 (13-00169-02) PCR2		KP738162		
					YHV7 (13-00169-03) PCR2		KP738163		
					YHV7 (13-00169-02) PCR3		KP738164		
		YHV (genotype 8)	China	*Fenneropenaeus chinensis*	20120706	Complete genome	KX947267	2012	[101]
		TSV	Ecuador	Penaeid shrimp	EC1993a	Capsid protein 2 gene	FJ876460	1993	[127]
					EC1993b		FJ876461		
					EC1994		FJ876466	1994	
					EC2006a		FJ876512	2006	
					EC2006b		FJ876513		
			Columbia	Penaeid shrimp	CO1994a		FJ876462	1994	
					CO1994b		FJ876463		
					CO1994c		FJ876464		
					CO1994d		FJ876465		
					CO1998		FJ876477	1998	
				*Penaeus vannamei*	CO-06A		JN194141	2006	[138]
					CO-06B		JN194142		
					CO-06C		JN194143		
					CO-07A		JN194144	2007	
					CO-07B		JN194145		
					CO-10		JN194146	2010	
					CO10	Complete genome	JF966384	2010	
			USA	*Penaeus vannamei*	94USHI	Complete genome	AF277675	1994	[62,132,194,195]
					HI94TSV	Viral coat protein 2 gene	AY826054	1994	[117]
						Viral coat protein 3 gene	AY826055		
					US-TX04	Complete genome	GQ502201	2004	[132]
					2005-334		MT877007	2019	[119]
				Penaeid shrimp	US1994	Capsid protein 2 gene	FJ876468	1994	[127]
					US1995		FJ876469	1995	
					US1996		FJ876474	1996	
					US1998		FJ876476	1998	
					US2004		FJ876492	2004	
					US2007		FJ876517	2007	
			Honduras	Penaeid shrimp	HO1994	Capsid protein 2 gene	FJ876467	1994	
					HO1998		FJ876475	1998	
					HO2003		FJ876483	2003	
			Mexico	Penaeid shrimp	MX1995a	Capsid protein 2 gene	FJ876470	1995	[127]
					MX1995b		FJ876471		
					MX1995c		FJ876472		
					MX1996		FJ876473	1996	
					MX1998		FJ876478	1998	
					MX1999a		FJ876479	1999	
					MX2000		FJ876480	2000	
					MX2004		FJ876493	2004	
					MX2005a		FJ876504	2005	
					MX2005b		FJ876505		
					MX2005c		FJ876506		
					MX2006		FJ876514	2006	
					MX2007		FJ876521	2007	
				*Penaeus vannamei*	SIN98TSV	Viral coat protein 1 gene	AF510515	1998	[125,195]
					MX99	Coat protein gene	AF277378	1999	[126,127]
					Mexico 10	Capsid protein 2 gene	JN194147	2010	[138]
				*Penaeus stylirostris*	MX99TSV	Viral coat protein 1 gene	AF510516	1999	[125,195]
					SON2KTSV		AF510517	2000	[131,195]
				*Penaeus stylirostris*	HI94TSV	Viral coat protein 1 gene	AF510518	2000	[117,125]
			Taiwan	*Penaeus vannamei*	TW99	Coat protein gene	AF406789	1999	[62,126,195]
				*Penaeus monodon*	Tw2KPmTSV	Capsid protein precursor	AY355309	2000	[126]
				*Metapenaeus ensis*	Tw2KMeTSV	Capsid protein precursor	AY355310	2000	[196]
				*Penaeus vannamei*	Tw02PvTSV	Capsid protein precursor	AY355311	2002	[127]
				Penaeid shrimp	TW2007	Capsid protein 2 gene	FJ876520	2007	
			Thailand	*Penaeus vannamei*	Th03-1TSV	Capsid protein 2 gene	DQ000304	2003	[196]
					Th03-2TSV		DQ000305		
					ThOct03LvTSV	VP1 gene	AY912503	2003	[126]
					ThMar04LvTSV		AY912504	2004	
					ThJul04LvTSV		AY912508		
				*Penaeus monodon*	ThMar04Pm1TSV	VP1 gene	AY912505	2004	
					ThMar04Pm2TSV		AY912506		
				*Penaeus monodon* (post-larvae)	ThMay04PmPLTSV	VP1 gene	AY912507	2004	
				*Penaeus vannamei*	TH03-1	Capsid protein 1 gene	AY755587	2003	[125,196]
					TH03-2		AY755588		
					TH03-3		AY755589		
					TH03-4		AY755590		
					TH03-5		AY755591		
					TH03-7		AY755593		
					TH03-9		AY755595		
					TH04Lv	Complete genome	AY997025	2005	[132,197]
				*Macrobrachium rosenbergii*	TH03-6	Capsid protein 1 gene	AY755592	2003	[125]
				*Penaeus monodon*	TH04Pm	Capsid protein 2 gene	DQ000306	2004	[196]
					TH03-8	Capsid protein 1 gene	AY755594	2003	[125]
				Penaeid shrimp	TH2003a	Capsid protein 2 gene	FJ876484	2003	[127]
					TH2003b		FJ876485		
					TH2004a		FJ876496	2004	
					TH2004b		FJ876497		
					TH2006		FJ876515	2006	
			Myanmar	*Penaeus monodon*	Mm03Pm	Capsid protein 1 gene	AY755596	2003	[125,196]
			Vietnam	*Penaeus vannamei*	VN-TSV	Capsid protein gene	AY694136	-	[198]
			Belize	*Penaeus vannamei*	BZ01	Non-structural polyprotein gene; Capsid protein precursor gene	AY590471	2001	[62,124,132]
				*Penaeus vannamei*	2005-175	Complete gene	MT877008	2019	[119]
					BLZ02TSV	Viral coat protein 1 gene	AY826051	2002	[117]
						Viral coat protein 2 gene	AY826052		
						Viral coat protein 3 gene	AY826053		
				Penaeid shrimp	BH2001	Capsid protein 2 gene	FJ876481	2001	[127]
					BH2002		FJ876482	2002	
					BH2004a		FJ876490	2004	
					BH2004b		FJ876491		
					BH2005a		FJ876498	2005	
					BH2005b		FJ876499		
					BH2005c		FJ876500		
					BH2008		FJ876522	2008	
			Indonesia	*Penaeus vannamei*	Id03TSV	Capsid protein 2 gene	DQ000303	2003	[196]
				*Penaeus vannamei*	Indonesia 10		JN194148	2010	[138]
				Penaeid shrimp	ID2003a		FJ876486	2003	[127]
					ID2003b		FJ876487		
					ID2003c		FJ876488		
					ID2005		FJ876501	2005	
					ID2006		FJ876510	2006	
			China	*Penaeus vannamei*	ZHZC3TSV	Complete genome	DQ104696	2005	[132,199]
					Cn03TSV	Capsid protein 2 gene	DQ000301	2003	[196]
					Ch-1	Capsid protein 1 gene	AY755597		[125,200]
					Ch-2		AY755598		
					Ch-3		AY755599		
					Ch-4		AY755600		
					Ch-6		AY755602		
				*Penaeus japonicus*	Ch-5	Capsid protein 1 gene	AY755601	2003	[125]
				Penaeid shrimp	CH2003a	Capsid protein 2 gene	FJ876489	2003	[127]
					CH2004		FJ876494	2004	
					CH2005a		FJ876509	2005	
					CH2007		FJ876518	2007	
			Korea	*Penaeus vannamei*	KOR-CsPv04TSV	Capsid protein 1 mRNA	DQ099912	2004	[131]
					KOR-ImPv05TSV		DQ099913		
			Eritrea	*Penaeus monodon*	Er04PmTSV	Capsid protein 2 gene	DQ000302	2004	[196]
				Penaeid shrimp	ER2004		FJ876495	2004	[127]
			Venezuela	*Penaeus vannamei*	VE05	Complete genome	DQ212790	2005	[124]
					2005-194		MT877006	2019	[119]
				Penaeid shrimp	VE2005a	Capsid protein 2 gene	FJ876502	2005	[127]
					VE2005b		FJ876503		
			Saudi Arabia	Penaeid shrimp	SA2007	Capsid protein 2 gene	FJ876519	2007	
				*Penaeus indicus*	SAPi	Complete genome	JX094350	2010	[118]
					SA2010a	Capsid protein 2 gene	JQ356858		
					SA2010b		JQ356859		
					SA2010c		JQ356860		
					SA2011a		JQ356861	2011	
					SA2011b		JQ356862		
					SA2011c		JQ356863		
					SA2011d		JQ356864		
					SA2011e		JQ356865		
			Aruba	Penaeid shrimp	AW2005	Capsid protein 2 gene	FJ876508	2005	[127]
					AW2006		FJ876511	2006	
			Nicaragua	Penaeid shrimp	NI2005	Capsid protein 2 gene	FJ876507	2005	
					NI2006		FJ876516	2006	
		WTD (*Mr*NV)	French West Indies	*Macrobrachium rosenbergii*	*Mr*NV	Segment RNA-1	AY222839	2003	[143,154]
						Segment RNA-2	AY222840		
						RNA-1	NC_005094	2009	[201]
						RNA-2	NC_005095	-	
					*Mr*NV-Ant	Putative RNA-dependent RNA-polymerase gene	AY313773	2005	[141]
			China		*Mr*NV	RNA-directed RNA polymerase gene	AAQ54758	-	[202]
					Chinese 1		AY231436	2006	[143,202]
					Chinese 2	Segment RNA-2	FJ751225	-	[143]
					*Mr*NV	Segment RNA-1 RNA-dependent RNA polymerase gene; B2 protein gene	FJ751226	2006	[201]
						Capsid protein gene	AY231437	-	[143]
			India		Nellore	Capsid protein gene	GU300102	-	[203]
						B2 protein gene	GU300103	2011	
					*Mr*NV	Capsid protein-like gene	HM565741	2010	[143]
						RNA-1 RNA-dependent RNA polymerase gene; B2 protein gene	JQ418295	-	[153]
						RNA-2 capsid protein gene	JQ418298	-	[149,200]
						Capsid protein	AM114036	-	
						RNA-dependent RNA polymerase gene	AAO60068	-	[152]
						RNA-directed RNA polymerase gene	DQ146969	-	[201]
					Kakinada 1*Mr*NV	Isolate Kakinada 1MrNV capsid protein gene	HQ637179	2008	[149]
			Taiwan		AC06-016	RNA-directed RNA polymerase gene	DQ459203	-	[143]
					AC06-017		DQ459204		
					AC06-024		DQ459205		
					AC06-86		DQ459206		
					AC06-088		DQ459207		
					AC06-89		DQ459208		
					*Mr*NV	Segment RNA-1 nonfunctional polymerase gene	DQ521574	-	[201]
						Segment RNA-2 capsid protein gene	DQ521575	-	
			Malaysia		*Mr*NV	Dependent RNA polymerase gene	JN187416	2009	[143]
			Australia		07-265.1	Capsid protein gene	FJ379530	2007	[204]
					07-265.2	A protein gene	FJ379531		
					Australian	Segment RNA 1	JN619369	2004	[143]
						Segment RNA 2	JN619370		
			Thailand		M298	Capsid protein gene	EU150126	-	[143]
					M299		EU150127		
					M308		EU150128		
					M12		EU150129		
					*Mr*NV	Capsid protein mRNA	DQ189990	-	[201]
		WTD (XSV)	Taiwan	*Macrobrachium rosenbergii*	XSV	Nucleocapsid protein CP17 gene	DQ521573	-	[205]
			Thailand		M23	Capsid protein gene	EU150133	-	[204]
					M309		EU150132	-	
					07-265.3		FJ379532	2007	
			India		Kakinada 1XSV	Isolate Kakinada 1XSV capsid protein gene	HQ637180	2008	[149]
					XSV	Capsid protein gene	JQ418299	-	
						Capsid protein, genomic RNA	AM114037	-	
						Capsid protein gene	NC_043494	-	
						Capsid protein gene	AY247793	-	[198]
			China		XSV	Nucleocapsid protein CP17 and CP16 genes	DQ174318	-	[206]

**Table 4 viruses-14-00585-t004:** Summary of host species following DNA and RNA viral disease infections.

Type		Pathogen	Host Species	Characteristics	Reference
DNA virus	ds DNA	WSSV	*Penaeus monodon*	● White spots: - Diameter (1–2 mm) - Carapace, appendages, and inside surfaces - Cuticle of cephalothorax and tail part - Calcium deposition on the inner surface of cuticle ● Lethargic ● Reddish body discoloration - Pleopods - Periopods - Telson - Uropods ● Discoloration of the hepatopancreas ● Loss of appetite ● Reduced swimming activity ● Reduced preening activity ● Disorientation during swimming ● Loosening of the cuticle ● Branchiostegites swelling ● Thinning and delayed clotting of the hemolymph ● Reduction of food consumption ● Gathered near the pond edge ● Tendency to move towards the edges of tanks, near the surface	[9,13,18,20,44,49,51,53,207,208,209,210]
			*Penaeus indicus*		
			*Penaeus japonicas*		
			*Penaeus chinensis*		
			*Penaeus penicillatus*		
			*Penaeus semisulcatus*		
			*Penaeus aztecus*		
			*Penaeus vannamei*		
			*Penaeus merguiensis*		
			*Penaeus duorarum*		
			*Penaeus stylirostris*		
			*Trachypenaeus curvirostris*		
			*Metapenaeus ensis*		
			*Exopalaemon orientalis*		
			*Macrobrachium rosenbergii*		
			*Marsupenaeus japonicus*		
			*Metapenaeus dobsoni*		
			*Parapenaeopsis stylifera*		
			*Solenocera indica*		
			*Squilla mantis*		
			*Procambarus clarkii*	● Loss of appetite ● Lethargy ● White spots on the carapace ● Loosening of the stratum corneum ● Discoloration of the hepatopancreas ● White calcification spots on the exoskeleton ● Dark coloration on the dorsal side ● Reduced swimming	[48,50,207,211]
			*Pacifastacus leniusculus*		
			*Orconectes punctimanus*		
			*Austropotamobius pallipes*		
			*Panulirus versicolor*	● Lack of appetite ● Dark coloration on the dorsal side ● Reduced swimming activity ● Lack of movement ● Not observed white spots	[212,213]
			*Panulirus penicillatus*		
			*Panulirus homarus*		
			*Panulirus ornatus*		
			*Charybdis feriatus*	● Reduced swimming activity ● Degenerated cells ● Lack of movement ● Lack of appetite ● Lethargy ● Basophilic intranuclear inclusions of the: - Gill - Head muscle - Muscle - Eyestalks - Heart tissue ● Dark and pinkish color on the dorsal side	[48,50,207,210,212,214]
			*Charybdis cruciata*		
			*Portunus pelagicus*		
			*Portunus sanguinolentus*		
			*Charybdis granulata*		
			*Scylla serrata*		
			*Helice tridens*		
			*Carcinus maenas*		
			*Calappa lophos*		
			*Paratelphusa hydrodomous*		
			*Paratelphusa pulvinata*		
			*Matuta planipes*		
	ss DNA	IHHNV	*Penaeus vannamei*	● Target organs: - Ectodermal (cuticular epidermis, hypodermal epithelium of the fore and hind gut, nerve cord and nerve ganglia) - Mesodermal (hematopoietic organs, antennal gland, tubule epithelium, gonads, lymphoid organ, connective tissue and striated muscles) - Origin (i.e., hepatopancreas, midgut epithelium, anterior mid-gut caecum or posterior midgut caecum) ● Acute infection: - Post-larvae and juveniles - Stop swimming - Tumble - Slowly sink to the bottom of the pond ● Chronic infection: - Juvenile and subadult - Growth retardation - Deformed rostrum - Wrinkled antennal flagella - Cuticular roughness ● Susceptible: - All life stages - Sensitive stage: Larvae and juvenile - Carrier stage: Adults - Low mortality: *Penaeus vannamei*	[57,58,59,66,85]
			*Penaeus stylirostris*		
			*Penaeus occidentalis*		
			*Penaeus monodon*		
			*Penaeus semisulcatus*		
			*Penaeus californiensis*		
			*Penaeus schmitti*		
			*Penaeus japonicus*		
			*Penaeus latisulcatus*		
			*Penaeus chinensis*		
			*Penaeus setiferus*		
			*Penaeus aztecus*		
			*Penaeus duorarum*		
			*Penaeus subtilis*		
			*Artemesia longinaris*		
			*Macrobrachium rosenbergii*		
			*Palaemon macrodactylus*		
			*Procambarus clarkii*		
			*Hemigrapsus penicillatus*		
			*Neohelice granulate*		
			*Corydoras arcuatus*		
			*Mytilus edulis*		
			*Mactra chinensis*		
			*Tegillarca granosa*		
			*Ruditapes philippinarum*		
			*Sinonovacula constricta*		
			*Meretrix meretrix*		
			*Mactra veneriformis*		
RNA virus	ds RNA	IMNV	*Penaeus vannamei*	● Target tissue - Skeletal muscles - Gills and lymphoid organ ● Acute infection: - Clinical manifestation is prominent - Moribund - Lethargy during or soon after stressful events such as netting, feeding, sudden changes in water temperature and sudden reductions in water salinity - Extensive necrotic areas in skeletal muscle tissues - Distal abdominal segments - White and opaque tail muscle - Milky tail - Pink hue of tail ● Chronic infection: - Liquefying of the necrotic muscles - Reddish coloration of the muscles and appendices ● Susceptible - Occur at any stage - Most susceptible stage: Juvenile	[62,73,76,83,84,85,86,195]
			*Penaeus stylirostris*		
			*Penaeus monodon*		
			*Farfantepenaeus subtiltis*		
	ss RNA	YHD	*Penaeus stylirostris*	● Necrosis: - Lymphoid organ - Gills - Connective tissues - Hemocytes - Hematopoietic organs ● Hepatopancreas - Yellow coloration - Atrophy - Soft ● Faded body color ● Yellow coloration of the cephalothorax and gills ● Congregate at pond edges near the surface ● Irregular swimming pattern ● Cessation of feeding	[68,91,100,101,106,113,215]
			*Penaeus aztecus*		
			*Penaeus duorarum*		
			*Penaeus setiferus*		
			*Penaeus vannamei*		
			*Penaeus esculentus*		
			*Penaeus stylirostris*		
			*Penaeus monodon*		
			*Fenneropenaeus merguiensis*		
			*Farfantepenaeus aztecus*		
			*Farfantepenaeus duorarum*		
			*Metapenaeus ensis*		
			*Metapenaeus affinis*		
			*Marsupenaeus japonicus*		
		TSV	*Penaeus stylirostris*	● Acute infection - Reddish body color, especially on the tail - Red chromatophore expansion - Irregular black spot under the cuticle layer - Lethargy - Anorexia - Opaque musculature - Flaccid bodies - Soft cuticle ● Transitional infection - Multifocal melanized lesions of the cephalothorax and tail - Lethargy - Anorexia ● Chronic infection - Cessation of mortality - Absence of disease signs - Resumption of normal feeding and swimming behavior	[52,95,129,130,136,216]
			*Penaeus schmitti*		
			*Penaeus setiferus*		
			*Penaeus duorarum*		
			*Penaeus aztecus*		
			*Penaeus monodon*		
			*Penaeus japonicus*		
			*Penaeus chinensis*		
		WTD	*Macrobrachium rosenbergii*	● Clinical signs: - Lethargy - Opaqueness of the abdominal muscle - Degeneration of the telson and uropods ● Susceptible stages: - Hatchery and nursery phases - Larvae - Post-larvae - Juvenile ● Carrier stage - Adult	[56,140,144,160,163]
			*Penaeus indicus*		
			*Penaeus japonicus*		
			*Penaeus monodon*		
			*Penaeus vannamei*		
			*Cherax quadricarinatus*		

**Table 5 viruses-14-00585-t005:** Summary of the DNA and RNA viral diseases PCR analyses.

Type		Pathogen	PCR	Host	Tissue	Primer	Sequence 5′-3′		Annealing Temperature (°C)	Amplicons (bp)	Reference
DNA virus	ds DNA	WSSV	Conventional PCR	*Cherax quadricarinatus*; *Procambarus clarkii*	Hepatopancreas, gills, cuticle, muscle	WSI3	GTA ACT CCT TCC ATC TCC A		62	941	[217]
						WSI4	TAC GGC AGC TGC TGC ACC TTG T				
				*Penaeus monodon*	Muscle	WSSV-VP28 F	TGT GAC CAA GAC CAT CGA AAC		52	516	[27]
						WSSV-VP28 R	TCG GTC TCA GTG CCA GAG TA				
			Real-time qPCR (EVA green)	*Penaeus vannamei*	Gills	VP24 F1	AGG ACC CGA TCG CTT ACT TTG		-	240	[218]
						VP24 R1	CTC CCT CCC TTG CGA ACT T				
						β-Actin F1	GAA GTA GCC GCC CTG GTT G			416	
						β-Actin R1	CGG TTA GCC TTG GGG TTG AG				
			Real-time PCR (BRYT Green)	*Penaeus monodon*	Muscle	WSSV-qVP28 F	TGT GAC CAA GAC CAT CGA AA		53	148	[27]
						WSSV-qVP28 R	CTT GAT TTT GCC CAA GGT GT				
			Real-time PCR (TaqMan)	*Cherax quadricarinatus*; *Procambarus clarkii*	Hepatopancreas, gills, cuticle, muscle	WSS1011F	TGG TCC CGT CCT CAT CTC AG		60	69	[217]
						WSS1079R	GCT GCC TTG CCG GAA ATT A				
			Nested PCR	*Fenneropenaeus indicus*	Pleopod	146F1	First	ACT ACT AAC TTC AGC CTA TCT AG	55	1447	[150]
							Second	GTA ACT GCC CCT TCC ATC TCC A		941	
	ss DNA	IHHNV	Conventional PCR	*Penaeus monodon*	Tissues of infected samples	77012F	ATC GGT GCA CTA CTC GGA		53	356	[58]
						77353R	TCG TAC TGG CTG TTC ATC				
				*Penaeus vannamei*		IHHNV389F	CGG AAC ACA ACC CGA CTT TA		55	389	
						IHHNV389R	GGC CAA GAC CAA AAT ACG AA				
						IHHNV392F	GGG CGA ACC AGA ATC ACT TA			392	
						IHHNV392R	ATC CGG AGG AAT CTG ATG TG				
				*Penaeus stylirostris; Penaeus vannamei*		IHHNV721F	TCT ACT GCC TCT GCA ACG AG			2000	
						IHHNV2860R	GTG GGT CTG GTC CAC TTG AT				
				*Penaeus monodon*		IHHNV3065F	GAC GAC GAA GAA TGG ACA GA			3000	
						IHHNV3065R	TGC CTG GGT AGC TGG TAT GTA TA				
						IHHNV309F	TCC AAC ACT TAG TCA AAA CCA A			309	
						IHHNV309R	TGT CTG CTA CGA TGA TTA TCC A				
				*Penaeus vannamei*	Hepatopancreas	IHHNV REPF	CGA TGT GCA ATA TAT ACC CGA TT		52	442	[57]
						IHHNV REPR	CTT CGC AGA AAC CGT TAA CTT				
						IHHNV472F	ACG AAC GAC CAC CCA TGG CA		57	472	
						IHHNV472R	TCT GGT TCG CCC TGA CGT GT				
						IHHNV447F	CGA AGC GCG AGT ATC CAT CA		55	447	
						IHHNV447R	TGA GTG ATG GAC GAA AGC GG				
						IHHNV-F	TCA TGA AGC GCG AGT ATC CAT CAT		54	228	
						IHHNV-R1	TGG GTG GTC GTT CGT ATC TT				
			Real-time PCR (TaqMan)	*Penaeus monodon*	Gills	IHHNV-q309F1	CCT AAA GAA AAC AGT GCA GAA TAT GAC		60.7	98	[219]
						IHHNV-q309R1	TCA TCG TCA AGT TTA TTG ACA AGT TC		60.8		
						IHHNV-qEVEF1	CCC ACA AAA AGC AAA TAT ATC TCA CTA T		61.1	106	
						IHHNV-qEVER1	GTC ATT ATG AGA TTA TTG TCC CAC CTT		61.7		
						Pmon-EF1qF1	GGC CGT GTG GAG ACT GGT AT		62.3	110	
						Pmon-EF1qR1	CGT GGT GCA TCT CCA CAG A		62.0		
			Real-time PCR (SYBR Green)	*Penaeus vannamei*	Gillsm muscle, hepatopancreas, hemolymph	IHHNV 195F	GGG AGT TAC CTT TGC TGC		56	195	[220]
						IHHNV 195R	GGT CCG TCT ACT GCG TCT				
RNA virus	ds RNA	IMNV	Reverse transcriptase PCR	*Penaeus vannamei*	Muscle	389F	CGG AAC ACA ACC CGA CTT TA		55	284	[62]
						389R	GGC CAA GAC CAA AAT ACG AA				
				*Penaeus vannamei*	Muscle	IMNV _105-297_-F	CAT ATG GGG CAA TTA CGG TTA CAG GG		60	600	[74]
						IMNV _105-297_-R	CGG GAT CCG TAT ACA TAC CAA ATG GCC				
						IMNV _300-527_-F	CTC GAG ACT AAA CAA ACA ACA GAC AAT GC		55	700	[87]
						IMNV _300-527_-R	GGA TCC GGA GTC CCA TCA TAT AAC TGG				
						IMNVF22	C CAT ATG ATT GTT TCA ATG GAA AAT C		57	811	[84]
						IMNVR819	G GAA TTC TTG TAG TGC AGT TGC TGG				
						IMNVF820	CGG GA TCC GCT GCA AAA GAG GGT GCT CG			924	
						IMNVR1728	G GAA TTC TTG CAT TGA ACTCCACGAAAA C				
						IMNVF1729	CG GGA TCC GGT AGT ATT GCA CCA GCA ATG			1041	
						IMNVR	GGA ATT CTT ATA CTG TTG CTG T CG CTT G				
						IMNV 99372G09- F	CGA CGC TGC TAA CCA TAC A A		62	372	[221]
						IMNV 99372 G10-R	ACT CGC CTG TTC GAT CAA GT				
						IMNV-NF	GGC ACA TGC TCA GAG ACA		60	139	[89]
						IMNV-NR	AGC GCT GAG TCC AGT CTT G				
	ss RNA	YHD	RT-PCR	*Penaeus monodon*	Gills, hemolymph	YHV5f	CGT ATT GCA TCG AAC GTC ACT G		60	885	[222]
						YHV5r	CAA GAT CAC TAA TAA CGC CTG ATG C				
			Nested PCR			YHV2s	CGG GGT TAC CCG CTT ATA TT			400	
						YHV2as	GCC TGA GGT GAA GTC CAT GT				
			RT-PCR	*Penaeus monodon*	Gills, epidermis	YCF1a	ATC GTC GTC AGC TAC CGC AAT ACT GC		60	359	[98]
						YCF1b	ATC GTC GTC AGY TAY CGT AAC ACC GC				
						YCR1a	TCT TCR CGT GTG AAC ACY TTC TTR GC				
						YCR1b	TCT GCG TGG GTG AAC ACC TTC TTG GC				
			Nested PCR			YCF2a	CGC TTC CAA TGT ATC TGY ATG CAC CA		66	147	
						YCF2b	CGC TTY CAR TGT ATC TGC ATG CAC CA				
						YCR2a	RTC DGT GTA CAT GTT TGA GAG TTT GTT				
						YCR2b	GTC AGT GTA CAT ATT GGA GAG TTT RTT				
			Real time RT-qPCR (TaqMan)	*Penaeus monodon*	Pleopod	GAVQPF1	GGG ATC CTA ACA TCG TCA ACG T		60	-	[223]
						GAVQPR1	AGT AGT ATG GAT TAC CCT GGT GCA T				
						6FAM-TAMRA probe	6FAM-TCA GCC GCT TCC GCT TCC AAT G				
			RT-LAMP PCR	*Penaeus vannamei*	Pleopods	YHV-F3	ACC CTG TAA TTG GCG ATG TT		65	186	[113]
						YHV-B3	TGC AGT TAA GAT GGT CAC AG				
						YHV-FIP	AGA GCA CTG TAG ACT GGT GGG **TTT T**TG TGG AAC CTG AAG AAT GC				
						YHV-BIP-Biotin	Biotin-TCA GCA CCT GGG CTC GTC TC**T TTT** CGA CAG TGA TTG AAG ACT CG				
						YHV-LF	AAC TGT TGC AGA TCG GAT T				
						YHV-LB	ATG TGT CAT GAT ATT CTC				
						YHV FITC probe	CTC CAT CCA GAA A				
			YHV7-qPCR (TaqMan)	*Penaeus monodon*	Pleopods, gills	qYHV-F1	CAT CCA ACC TAT CGC CTA CA		-	79	[91]
						qYHV-F2	ACC TAT CGC CTA CAC AGC TA			73	
						qYHV-R1	TGT GAA GTC CAT GTG AAC GA			-	
						qYHV7-Pr1	6FAM- CAA CGA CAG ACA CCT CAT CCG TGA-BHQ1			-	
			YH7-PCR			YHV7-F1a	CCT ACA CGC ATG CTC TCT CTA TG		-	788	
						YHV7-R1b	GGT GTC TGT CGT TGT GTA TAG CT				
			YHV7-nPCR			YHV7-F2a	CAA ACA CCA ACC GAC ATT CAG T		58	412	
						YHV7-R2a	GCG ACA GTG CTT GAA GAC TTT AG				
		TSV	ConventionalPCR	*Penaeus monodon*	Gills, tail, body cuticles, swimming feet	9992F	AAG TAG ACA GCC GCG CTT		60	231	[129]
			Real-time RT-PCR (TaqMan)	Davidson’s-fixed paraffin-embedded (DFPE) shrimp tissue		TSV1004F	TTG GGC ACC AAA CGA CAT T		60	417	[119]
						TSV1075R	GGG AGC TTA AAC TGG ACA CAC TGT				
						TSV-P1	FAM-CAG CAC TGA CGC ACA ATA TTC GAG CAT C-TAMARA				
						TSV1004F	TTG GGC ACC AAA CGA CAT T			122	[120]
						TSV1075R	GGG AGC TTA AAC TGG ACA CAC TGT				
						TSV-probe	FAM-CAG CAC TGA CGC ACA ATA TTC GAG CAT C-TAMARA				
				*Penaeus vannamei*	Pleopods	TSV-55P1	GGC GTA GTG AGT AAT GTA GC		60	955	[116]
						TSV-55P2	CTT CAG TGA CCA CGG TAT AG				
			Real-time RT-PCR (SYBR green)	*Penaeus vannamei*	Cephalothorax	TSV-306F	CGT AAA TAG ACG GCC CAC AAA		60	79	[138]
						TSV384R	TGC ATC TAT ATA TCC AGG GAC TTA TCC				
						TSV-285F	TTC TAT AGG TCT GGT TTA AAA CGT AAA			232	
						TSV-516R	CGG TTT TCT CCA TCA TCG TT				
		WTD	Reverse transcriptase PCR	*Macrobrachium rosenbergii*	Infected sample	*Mr*-RdRp-F	GCA TTT GTG AAG AAT GAA CCG		50	729	[56]
						*Mr*-RdRp-R	CAT GTT CAACTTTCTCCACGT				
						q*Mr*NV-F	AGG ATC CAC TAA GAA CGT GG			211	
						q*Mr*NV-R	CACGGTCACAATCCTTGCG				
						*Mr*Nv2F	GAT ACA GAT CCA CTA GAT GAC C		55	681	
						*Mr*Nv2R	GAC GAT AGC TCT GAT AAT CC				
					Muscle	1A775	CCA CGT TCT TAG TGG ATC CT		55	850	[147]
						1B690	CGT CCG CCT GGT AGT TCC				
						*Mr*NV DBHF	ATG GCT AGA GGT AAA CAA AAT TC		50	564	[149]
						*Mr*NV DBHR	TCA TTG ATC ATC ACG CCT GAC A				
						*Mr*NV PEF	GGG CCG GAT CCA TGG CTA GAG GTA AAC AAA ATT C				
						*Mr*NV PER	GGC CAA GCT TTC ATT GAT CAT CAC GCC TGA CA				
					Infected sample	FL-XSV-F	CCA CGT CTA GCT GCT GAC GTT		50	796	[56]
						FL-XSV-R	AAG GTC TTT ATT TAT CGA CGC				
						XSV-F	GGA GAA CCA TGA GAT CAC G		55	507	
						XSV-R	CTG CTC ATT ACT GTT CGG AGT C				
						qXSV-F	AGC CAC ACT CTC GCA TCT GA		50	68	
						qXSV-R	CTC CAG CAA AGT GCG ATA CG				
					Muscle	XSV DBHF	ATG AAT AAG CGC ATT AAT AAT		50	525	[149]
						XSV DBHR	TTA CTG TTC GGA GTC CCA ATA				
						XSV PEF	GGG CCG GAT CCA TGA ATA AGC GCA TTA ATA AT				
						XSV PER	GGC CAA GCT TTT ACT GTT CGG AGT CCC AAT A				

## 4. Conclusions

In this review, we have looked at the DNA and RNA viral diseases affecting shrimp, which are listed by the World Organization for Animal Health. We have provided an overview of the basic characteristics of the viral disease pathogens that can be fatal to farmed shrimp, as well as the disease distribution range, information on the specific hosts, apparent clinical symptoms, disease transmission methods and vectors, mortality rates, diagnostic techniques, as well as strategies for control and prevention. The legal or illegal cross-border movement of living aquatic species for aquaculture has accelerated the spread of diseases and the demand for vaccines and therapeutics for their prevention. However, to find a fundamental solution, various studies on the etiology of these diseases are needed, and breeding organism-friendly aquaculture methods will be required, which consider animal welfare, such as maintaining an appropriate breeding density and a clean breeding environment, using SPF (specific pathogen free) or SPR (specific pathogen resistance), and nature-friendly breeding and nurturing for a disease-free and sustainable shrimp farming industry. The material in this review will help researchers and those working in the industry to better understand the major viral diseases of shrimp, and can be used as a basic data document to help prepare policy measures to prevent and control shrimp viral diseases in the future.

## Figures and Tables

**Figure 1 viruses-14-00585-f001:**
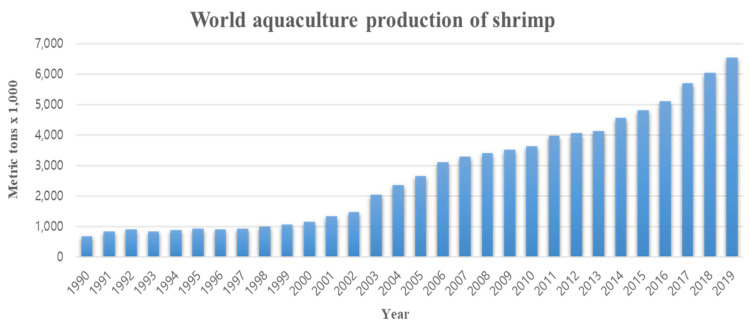
World aquaculture production of shrimp from 1990 to 2019 (Source: FAO yearbook of Fishery and Aquaculture Statistics).

**Figure 2 viruses-14-00585-f002:**
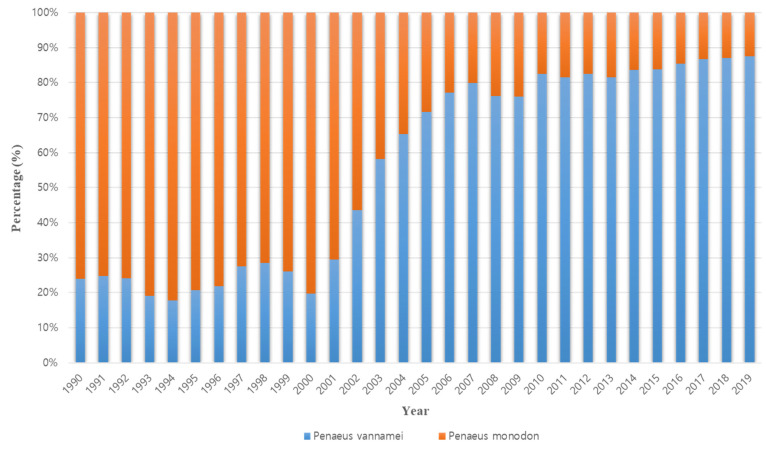
Proportion of the major shrimp species *Penaeus monodon* and *Penaeus vannamei* in aquaculture production from 1990 to 2019 (Source: FAO yearbook of Fishery and Aquaculture Statistics).

**Figure 3 viruses-14-00585-f003:**
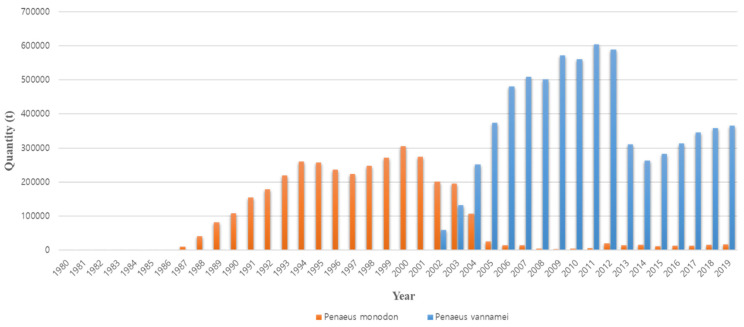
Total shrimp aquaculture production for *Penaeus monodon* and *Penaeus vannamei* in Thailand from 1980 to 2019 (Source: FAO Global Aquaculture Production Statistics from FishstatJ Software for Fishery and Aquaculture Statistical Time Series).

**Figure 4 viruses-14-00585-f004:**
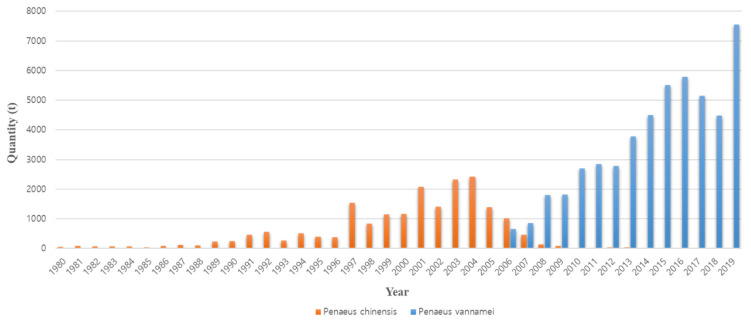
Total shrimp aquaculture production for *Penaeus chinensis* and *Penaeus vannamei* in the Republic of Korea from 1980 to 2019 (Source: FAO Global Aquaculture Production Statistics from FishstatJ Software for Fishery and Aquaculture Statistical Time Series).

**Figure 5 viruses-14-00585-f005:**
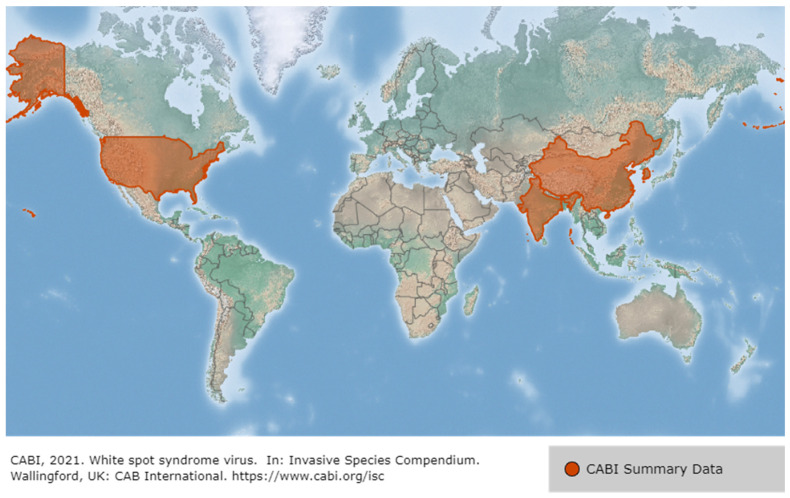
Distribution map showing the geographical occurrence of white spot syndrome disease (WSSD) (Reprinted from CABI, 2019, White spot syndrome virus. In: Invasive Species Compendium. Wallingford, UK: CAB International, with permission from CABI).

**Figure 6 viruses-14-00585-f006:**
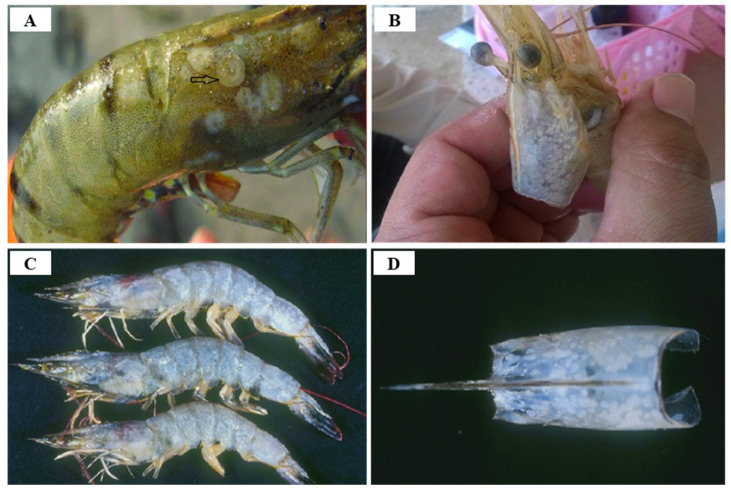
External white spot symptoms indicating white spot syndrome virus (WSSV) infection. (**A**) *Penaeus monodon* and (**B**–**D**) *Penaeus vannamei* infected with WSSV. (**A**) Reprinted from Letter in Applied Microbiology, Vol. 60 (2), Hossain, A., Nandi, S.P., Siddique, M.A., Sanyal, S.K., Sultana, M., Hossain, M.A., Prevalence and distribution of White Spot Syndrome Virus in cultured shrimp, p. 7, Copyright (2014), with permission from John Wiley and Sons; (**B**) Reprinted from Elsevier Books, Dashtiannasab, A., Emerging and Reemerging Viral Pathogens, p. 12, Copyright (2020), with permission from Elsevier; (**C**,**D**) Reprinted from Journal of Fish Diseases, Vol. 36 (12), Cheng, L., Lin, W.H., Wang, P.C., Tsai, M.A., Hsu, J.P., Chen, S.C., White spot syndrome virus epizootic in cultured Pacific white shrimp *Litopenaeus vannamei* (Boone) in Taiwan, p. 9, Copyright (2013), with permission from John Wiley and Sons).

**Figure 7 viruses-14-00585-f007:**
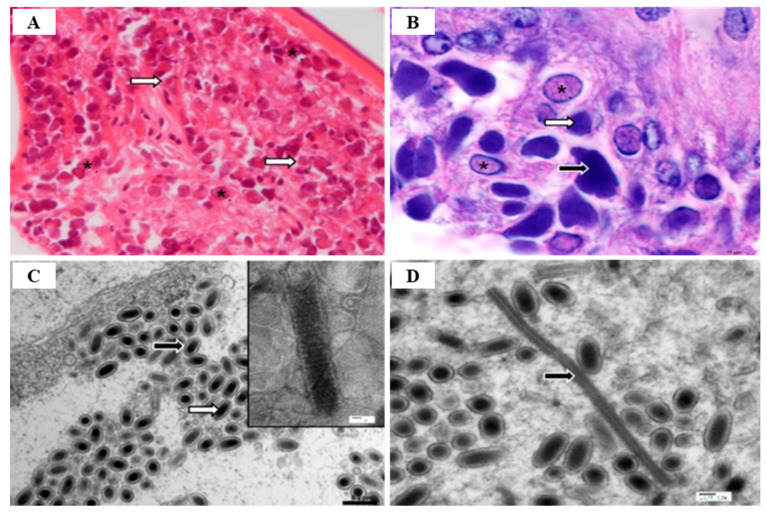
*Penaeus**vannamei* infected with white spot syndrome virus (WSSV). The infection progresses through different stages that can be seen in the nucleus via histology. (**A**) Early-stage infected cells display enlarged nuclei with marginalized chromatin and a homogenous eosinophilic central region. These then develop an intranuclear eosinophilic Cowdry A-type inclusion (*); this can be surrounded by a clear halo beneath the nuclear membrane (white arrow). Scale bar = 25 µm; (**B**) The eosinophilic inclusion usually expands to fill the nucleus (*). This inclusion becomes basophilic when staining and denser in color as the infection progresses (white arrow). Nuclei then disintegrate so that the content fuses with the cytoplasm (black arrow). Scale bar = 10 µm. H & E stain; (**C**) WSSV virions appear ovoid in shape and contain an electron-dense nucleocapsid (white arrow) within a trilaminar envelope (black arrow). Scale bar = 0.2 µm. Inset. Negatively stained WSSV nucleocapsid, showing the presence of cross-hatched or striated material that is structured as a series of stacked rings of subunits and is a key diagnostic feature of WSSV. Scale bar = 20 nm; (**D**) Presumptive nucleocapsid material within the nucleus prior to envelopment. This material is cross-hatched or striated in appearance and linear prior to its incorporation in the formation of mature WSSV particles. This linear nucleocapsid material is observed sporadically in the manufacture of the WSSV particles. Scale bar = 100 nm. Transmission electron microscopy images (Source: Verbruggen et al., 2016, https://doi.org/10.3390/v8010023 accessed on 11 May 2018).

**Figure 8 viruses-14-00585-f008:**
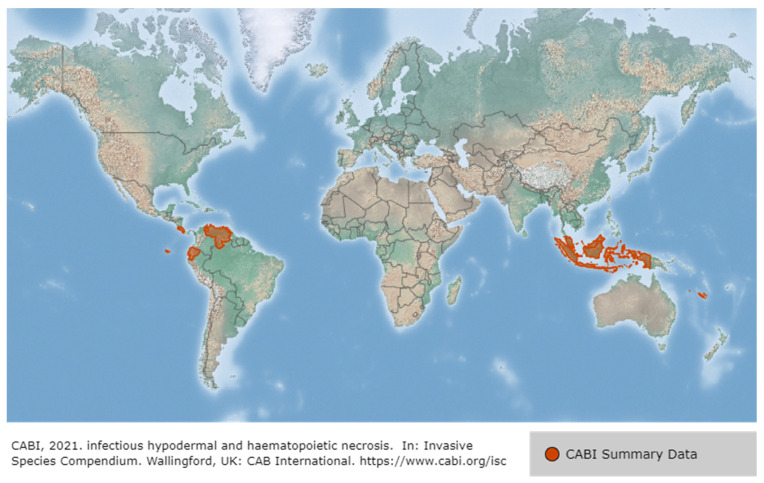
Distribution maps showing the geographical occurrence of infectious hypodermal and hematopoietic necrosis virus (Reprinted from CABI, 2019, Infectious hypodermal and hematopoietic necrosis. In: Invasive Species Compendium. Wallingford, UK: CAB International, with permission from CABI).

**Figure 9 viruses-14-00585-f009:**
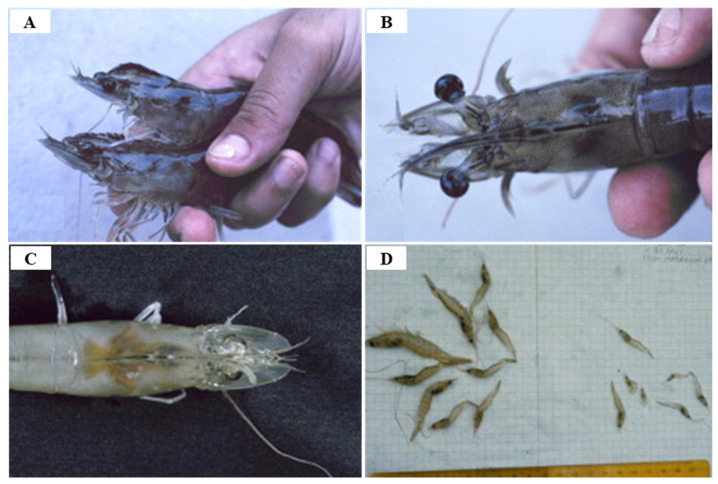
External symptoms of infectious hypodermal and hematopoietic necrosis virus (IHHNV) on shrimp. (**A**,**B**) subadult *Penaeus vannamei* with bent (to the left) rostrums, a classic sign of ‘runt deformity syndrome’ (RDS); (**C**) a juvenile *P. vannamei* with RDS. In this specimen the rostrum is bent to the right and the antennal flagella are wrinkled, brittle and mostly broken-off; (**D**) juvenile *P. vannamei* with RDS from a nursery population at approximately 60 days post stocking (Reprinted from Journal of Invertebrate Pathology, Vol. 106 (1), Lightner D.V., Virus diseases of farmed shrimp in the Western Hemisphere (the Americas) A rieview, p. 21, Copyright (2011), with permission from Elsevier).

**Figure 10 viruses-14-00585-f010:**
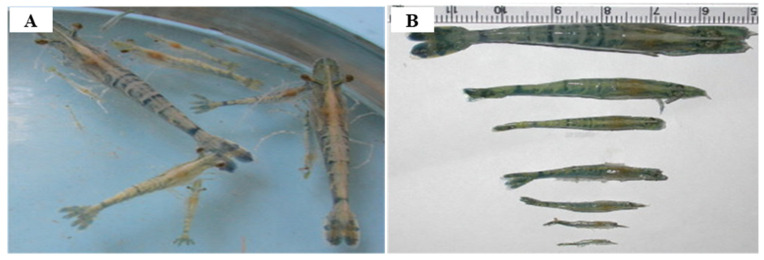
Size variations observed in 50-day-old *Penaeus monodon* with infectious hypodermal and hematopoietic necrosis virus (IHHNV) (**A**,**B**) (Reprinted from Aquaculture, Vol. 289 (3–4), Rai, P., Pradeep, B., Karunasagar, I., Karunasagar, I., Detection of viruses in *Penaeus monodon* from India showing signs of slow growth syndrome, p. 5, Copyright (2009), with permission from Elsevier).

**Figure 11 viruses-14-00585-f011:**
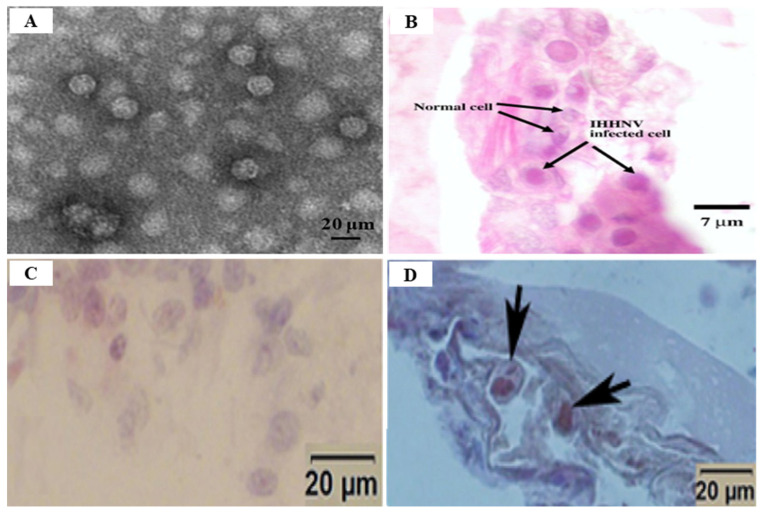
Electron microscopy and histological analysis of the changes in shrimp with infectious hypodermal and hematopoietic necrosis virus (IHHNV). (**A**) Electron microscopy of negatively stained IHHNV VLPs under self-assembly and disassembly conditions in *Penaeus vannamei*; (**B**) Cowdry type A eosinophilic inclusion of IHHNV in a nucleus of subcuticular epithelial cells of the pleopod of *P. monodon* (H & E, 1000×); (**C**) Histological detection of *Procambarus clarkii* gills negative to IHHNV detected by PCR. The gill cells were normal, no hypertrophied nucleus was observed; (**D**) Histological detection of *P. clarkii* gills positive to IHHNV detected by PCR. Several hypertrophied nuclei (arrow) were observed. ((**A**) Reprinted from Journal of Invertebrate Pathology, Vol. 166, Zhu, Y.P., Li, C., Wan, X.Y., Yang, Q., Xie, G.S., Huang, J., Delivery of plasmid DNA to shrimp hemocytes by infectious hypodermal and hematopoietic necrosis virus (IHHNV) nanoparticles expressed from a baculovirus insect cell system, p. 1, Copyright (2019), with permission from Elsevier; (**B**) Reprinted from Aquaculture, Vol. 289 (3–4), Rai, P., Pradeep, B., Karunasagar, I., Karunasagar, I., Detection of viruses in *Penaeus monodon* from India showing signs of slow growth syndrome, p. 5, Copyright (2009), with permission from Elsevier; (**C**,**D**) Reprinted from Aquaculture, Vol. 477, Chen, B.K., Dong, Z., Liu, D.P., Yan, Y.B., Pang, N.Y., Nian, Y.Y., Yan, D.C., Infectious hypodermal and hematopoietic necrosis virus (IHHNV) infection in freshwater crayfish *Procambarus clarkii*, p. 4, Copyright (2017), with permission from Elsevier).

**Figure 12 viruses-14-00585-f012:**
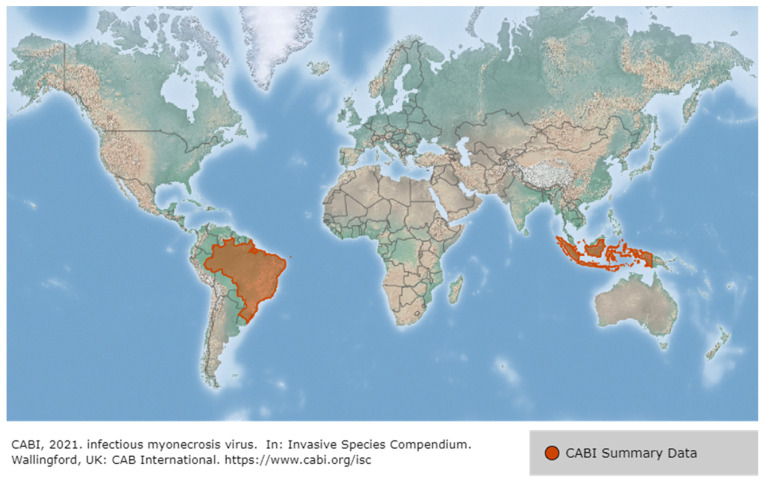
Distribution map showing the geographical occurrence of infectious myonecrosis virus (IMNV) (Reprinted from CABI, 2019, Infectious myonecrosis virus. In: Invasive Species Compendium. Wallingford, UK: CAB International, with permission from CABI).

**Figure 13 viruses-14-00585-f013:**
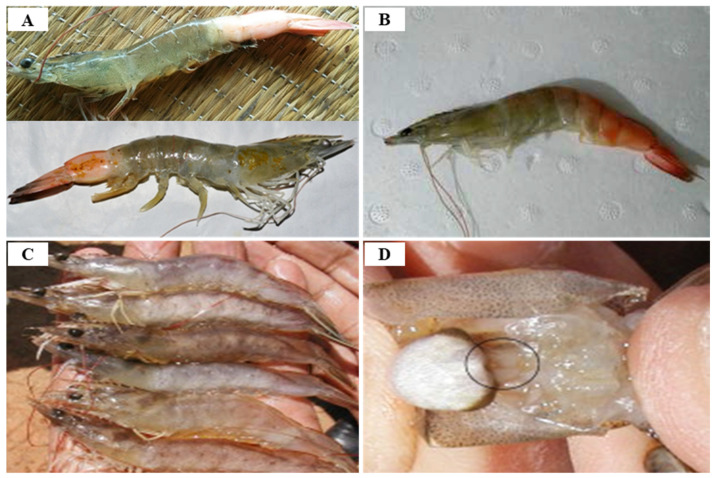
External symptoms of infectious myonecrosis virus (IMNV) on shrimp. (**A**) IMNV-infected *Penaeus vannamei* with reddish opaque muscles at the distal abdominal segments; (**B**) *P. vannamei* injected with IMNV propagated in a C6/36 cell line with reddish opaque muscle at the distal abdominal segments as observed in the natural infection; (**C**,**D**) *P. vannamei* infected with IMNV and displaying focal to extensive white necrotic areas in the striated muscle, especially of the distal abdominal segments and tail fan, and exposure of the paired lymphoid organs (LO) by simple dissection will show that the paired LO are hypertrophic to twice or more their normal size. ((**A**) Reprinted from Journal of Fish Diseases, Vol. 40 (12), Sahul Hameed, A.S., Abdul Majeed, S., Vimal, S., Madan, N., Rajkumar, T., Santhoshkumar, S., Sivakumar, S., Studies on the occurrence of infectious myonecrosis virus in pond-reared *Litopenaeus vannamei* (Boone, 1931) in India, p. 8, Copyright (2017), with permission from John Wiley and Sons; (**B**) Reprinted from Journal of Fish Diseases, Vol. 44 (7), Santhosh Kumar, S., Sivakumar, S., Abdul Majeed, S., Vimal, S., Taju, G., Sahul Hameed, A.S., In vitro propagation of infectious myonecrosis virus in C6/36 mosquito cell line, p. 6, Copyright (2021), with permission from John Wiley and Sons; (**C**,**D**) Reprinted from Journal of Invertebrate Pathology, Vol. 106(1), Lightner, D.V., Virus diseases of farmed shrimp in the Western Hemisphere (the Americas) a review, p. 21, Copyright (2011), with permission from Elsevier).

**Figure 14 viruses-14-00585-f014:**
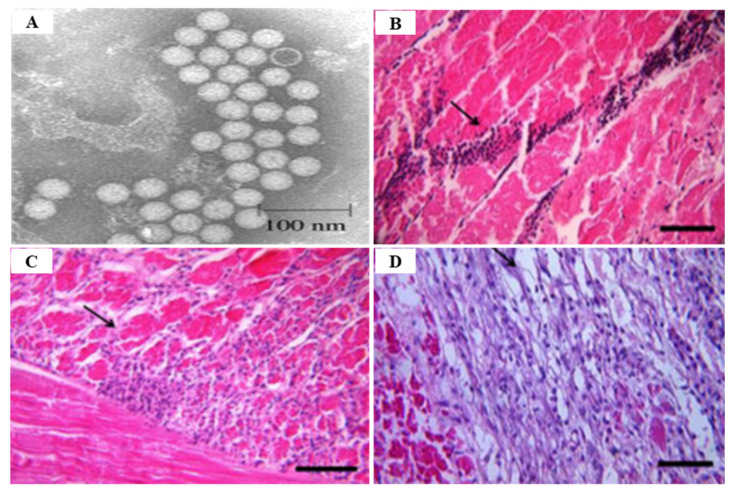
Electron microscopy and histological changes in shrimp with infectious myonecrosis virus (IMNV). (**A**) TEM of a purified preparation of IMNV from naturally infected *Penaeus vannamei* from Brazil. Photomicrographs of tissue sections from *P. vannamei* examined for IMNV lesions (**B**–**D**) (Scale bar = 50 μm); (**B**) Focal hemocytic infiltration in muscle tissue; (**C**) Muscle coagulation necrosis accompanied by infiltration of hemocytes; (**D**) Muscle liquefactive necrosis and fibrosis. ((**A**) Reprinted from Journal of Invertebrate Pathology, Vol. 106 (1), Lightner, D.V., Virus diseases of farmed shrimp in the Western Hemisphere (the Americas) a review, p. 21, Copyright (2011), with permission from Elsevier; (**B**–**D**) Reprinted from Aquaculture, Vol. 380, Feijó, R.G., Kamimura, M.T., Oliveira-Neto, J.M., Vila-Nova, C.M., Gomes, A.C., Maria das Graças, L.C., Maggioni, R., Infectious myonecrosis virus and white spot syndrome virus co-infection in Pacific white shrimp (*Litopenaeus vannamei*) farmed in Brazil, p. 5, Copyright (2013), with permission from Elsevier).

**Figure 15 viruses-14-00585-f015:**
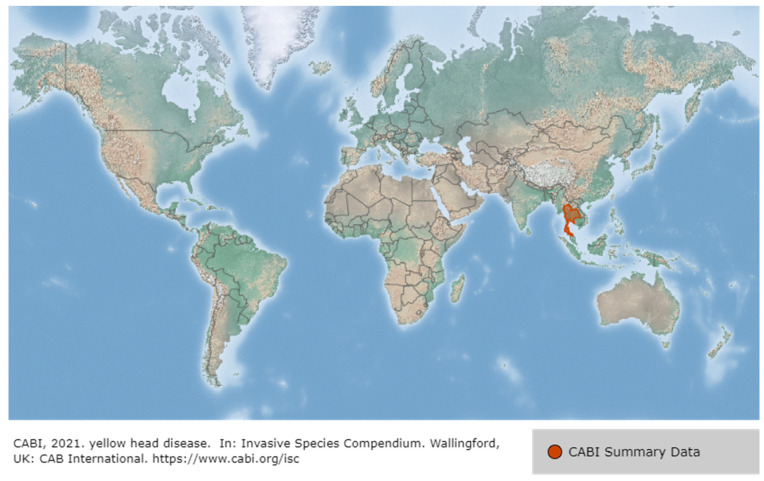
Distribution map showing the geographical occurrence of yellow head virus genotype 1 (YHV genotype 1) (Reprinted from CABI, 2019, Yellow head virus. In: Invasive Species Compendium. Wallingford, UK: CAB International, with permission from CABI).

**Figure 16 viruses-14-00585-f016:**
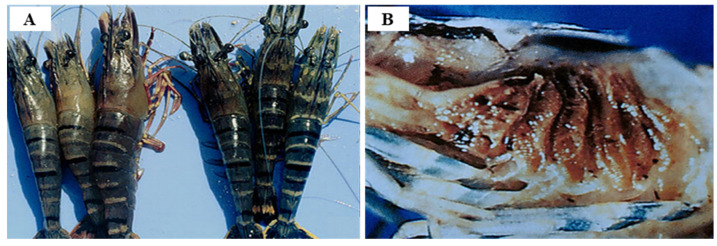
External symptoms on yellow head virus genotype 1 (YHV genotype 1)-infected shrimp. (**A**) *P. monodon* showing signs of yellow head disease (YHD) Yellow (light gray in print version) to yellow-brown (dark gray in print version) discoloration of the cephalothorax. Three shrimp with (left) and without (right) YHD; (**B**) discoloration of the gill region. ((**A**,**B**) Reprinted from Elsevier Books, Samocha, Sustainable biofloc systems for marine shrimp, p. 23, Copyright (2019), with permission from Elsevier).

**Figure 17 viruses-14-00585-f017:**
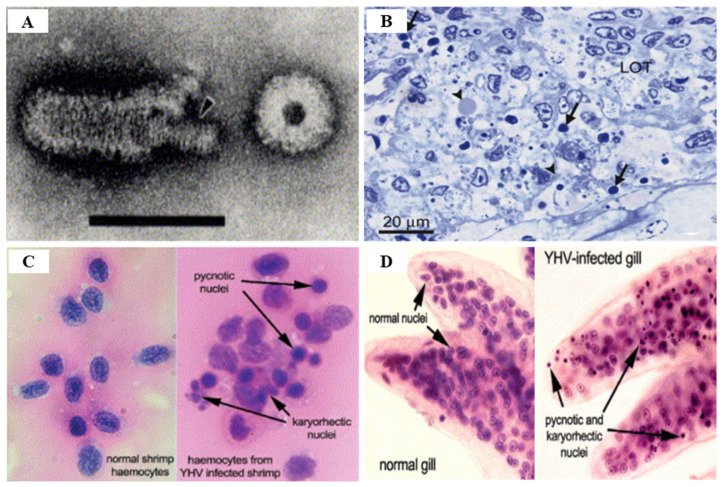
Electron microscopy and histological changes in shrimp infected with yellow head virus genotype 1 (YHV). (**A**) TEM of negative-strained YHV virions (Scale bars = 100 nm); (**B**) LO tissue of moribund shrimp from YHV immersion challenged *P. vannamei* at day 5 showing numerous pyknotic nuclei (arrows), karyorrhectic nucleic and cytoplasmic inclusion (arrow heads); (**C**) Hemolymph from normal and YHV infected shrimp identified by staining hemolymph smears; (**D**) Gills of YHV infected shrimp stained with H&E in rapidly fixed and stained (3 h) whole mounts. ((**A**) Reprinted from Advances in virus research, Vol. 63, Dhar, A.K., Cowley, J.A., Hasson, K.W., Walker, P.J., Genomic organization, biology, and diagnosis of Taura syndrome virus and yellow head virus of penaeid shrimp, p. 69, Copyright (2004), with permission from Elsevier; (**B**) Reprinted from Developmental & Comparative Immunology, Vol. 32 (6), Anantasomboon, G., Poonkhum, R., Sittidilokratna, N., Flegel, T.W., Withyachumnarnkul, B., Low viral loads and lymphoid organ spheroids are associated with yellow head virus (YHV) tolerance in whiteleg shrimp *Penaeus vannamei*, p. 14, Copyright (2008), with permission from Elsevier; (**C**,**D**) Reprinted from Aquaculture, Vol. 258 (1–4), Flegel, T.W., Detection of major penaeid shrimp viruses in Asia, a historical perspective with emphasis on Thailand, p. 33, Copyright (2006), with permission from Elsevier).

**Figure 18 viruses-14-00585-f018:**
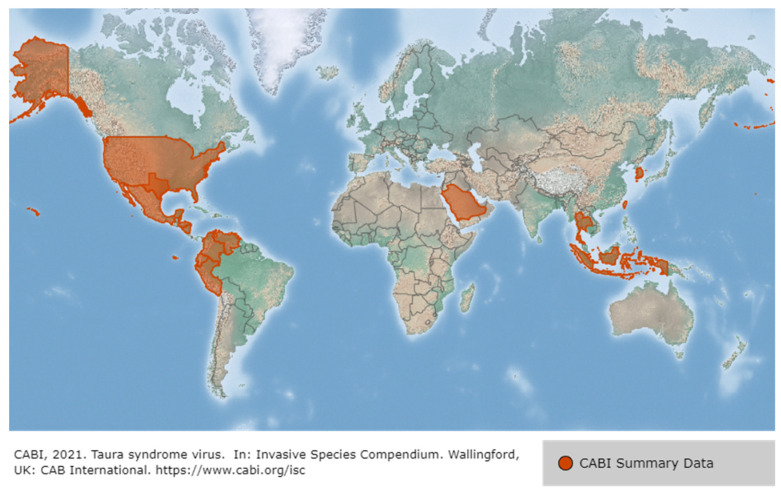
Distribution map showing the geographical occurrence of Taura syndrome virus (TSV) (Reprinted from CABI, 2019, Taura syndrome virus. In: Invasive Species Compendium. Wallingford, UK: CAB International, with permission from CABI).

**Figure 19 viruses-14-00585-f019:**
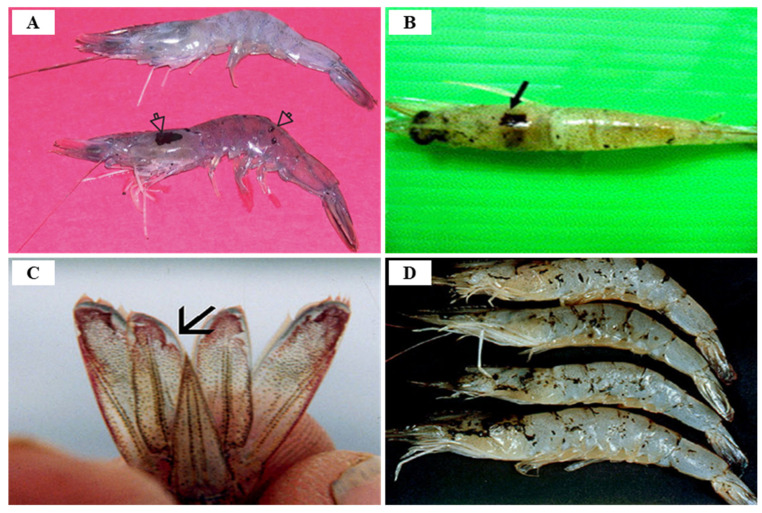
External symptoms of Taura syndrome virus (TSV) on infected shrimp. (**A**,**B**) *Penaeus vannamei* showing typical signs of TSV at the end of the acute phase: Multifocal and melanized lesions on the thorax and tail (indicated by arrow); (**C**,**D**) *P. vannamei* showing signs of TSV: red tail fan with rough edges on the cuticular epithelium of uropods (indicated by arrow) and multiple melanized cuticular lesions. ((**A**) Reprinted from Elsevier Books, Dhar, A.K., Allnutt, F.T., Taura Syndrome Virus. In Encyclopedia of virology, p. 8, Copyright (2008), with permission from Elsevier; (**B**) Reprinted from Aquaculture, Vol. 260 (1–4), Phalitakul, S., Wongtawatchai, J., Sarikaputi, M., Viseshakul, N., The molecular detection of Taura syndrome virus emerging with White spot syndrome virus in penaeid shrimps of Thailand, p. 9, Copyright (2006), with permission from Elsevier; (**C**,**D**) Reprinted from Elsevier Books, Samocha, Sustainable biofloc system for marine shrimp, p. 23, Copyright (2019), with permission from Elsevier).

**Figure 20 viruses-14-00585-f020:**
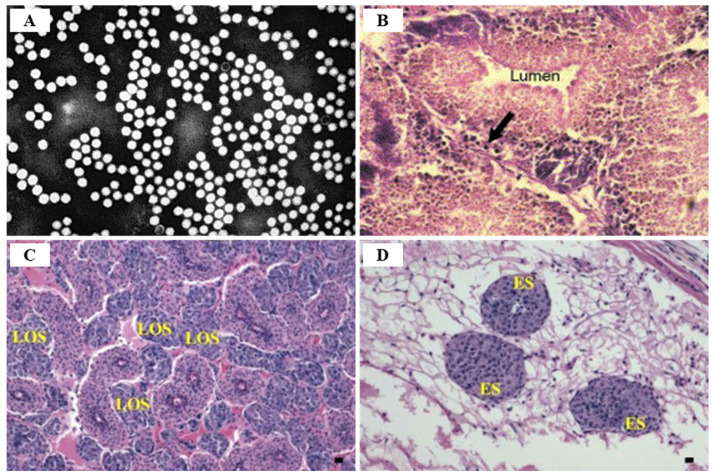
Electron microscopy and histological changes in shrimp infected with Taura syndrome virus (TSV). (**A**) TEM of CsCl gradient-purified and negative-strained (with 2% PTA) TSV particle isolated from *Penaeus vannamei* in Ecuador; (**B**) the section of intestine with 400 × magnification has cytoplasmic inclusion bodies in the lymphoid organ of *Penaeus monodon* (arrow); (**C**,**D**) spheroids (LOS) in the lymphoid organ tissue and ectopic spheroids in the connective tissue of *P. vannamei* from Venezuela, when stained with H&E, respectively (Scale bar = 25 μm). ((**A**) Reprinted from Advances in virus research, Vol. 63, Dhar, A.K., Cowley, J.A., Hasson, K.W., Walker, P.J., Genomic organization, biology, and diagnosis of Taura syndrome virus and yellowhead virus of penaeid shrimp, p. 69, Copyright (2004), with permission from Elsevier; (**B**) Reprinted from Aquaculture, Vol. 260 (1–4), Phalitakul, S., Wongtawatchai, J., Sarikaputi, M., Viseshakul, N., The molecular detection of Taura syndrome virus emerging with White spot syndrome virus in penaeid shrimps of Thailand, p. 9, Copyright (2006), with permission from Elsevier; (**C**,**D**) Reprinted from Aquaculture, Vol. 480, Tang, K.F., Aranguren, L.F., Piamsomboon, P., Han, J.E., Maskaykina, I.Y., Schmidt, M.M., Detection of the microsporidian Enterocytozoon hepatopenaei (EHP) and Taura syndrome virus in *Penaeus vannamei* cultured in Venezuela, p. 5, Copyright (2017), with permission from Elsevier).

**Figure 21 viruses-14-00585-f021:**
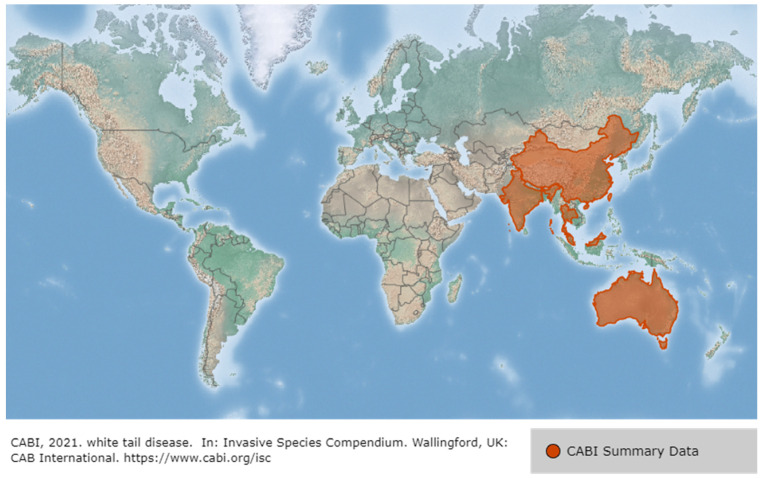
Distribution map of the geographical occurrence of White tail disease (WTD). (Reprinted from CABI, 2019, *Macrobrachium rosenbergii* nodavirus. In: Invasive Species Compendium. Wallingford, UK: CAB International, with permission from CABI).

**Figure 22 viruses-14-00585-f022:**
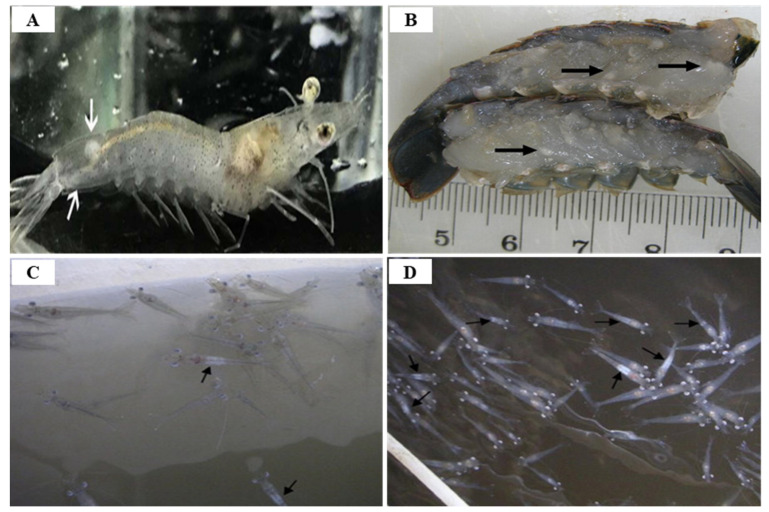
External symptoms of shrimps with White tail disease (WTD). (**A**) *Mr*NV-infected *Penaeus vannamei* showing signs of whitish muscle in the tail (arrows); (**B**) *Cherax quadricarinatus* showing signs of WTD with necrosis and myositis (arrows); (**C**,**D**) Clinical signs, whitish abdominal muscles (arrows), in the infected post-larvae of *Penaeus indicus* ((A) Reprinted from Aquaculture, Vol. 483, Jariyapong, P., Pudgerd, A., Weerachatyanukul, W., Hirono, I., Senapin, S., Dhar, A.K., Chotwiwatthanakun, C., Construction of an infectious *Macrobrachium rosenbergii* nodavirus from cDNA clones in Sf9 cells and improved recovery of viral RNA with AZT treatment, p. 9, Copyright (2018), with permission from Elsevier; (**B**) Reprinted from Aquaculture, Vol. 319 (1–2), Hayakijkosol, O., La Fauce, K., Owens, L., Experimental infection of redclaw crayfish (*Cherax quadricarinatus*) with Macrobrachium rosenbergii nodavirus, the aetiological agent of white tail disease, p. 5, Copyright (2011), with permission from Elsevier; (**C**,**D**) Reprinted from Aquaculture, Vol. 292(1–2), Ravi, M., Basha, A.N., Sarathi, M., Idalia, H.R., Widada, J.S., Bonami, J.R., Hameed, A.S., Studies on the occurrence of white tail disease (WTD) caused by *Mr*NV and XSV in hatchery-reared post-larvae of *Penaeus indicus and P. monodon*, p. 4, Copyright (2009), with permission from Elsevier).

**Figure 23 viruses-14-00585-f023:**
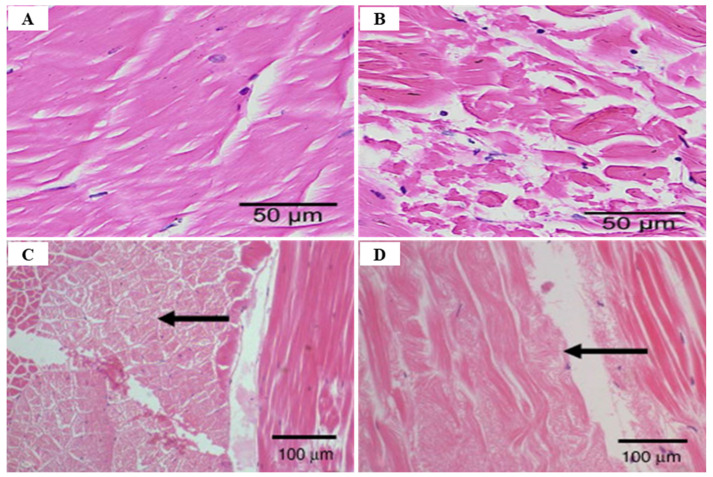
Histological changes in shrimp tissues when infected with White tail disease (WTD) and stained with H&E. (**A**) Uninfected shrimp; (**B**) Histological detection included the aggregation of cells into clumps of various sizes and coagulative necrosis in *P. vannamei* skeletal muscle (72 h post-infection); (**C**,**D**) Muscle degeneration and necrotic muscle tissues in *Mr*NV-infected *C. quadricarinatus* (arrow). ((**A**,**B**) Reprinted from Aquaculture, Vol. 483, Jariyapong, P., Pudgerd, A., Weerachatyanukul, W., Hirono, I., Senapin, S., Dhar, A.K., Chotwiwatthanakun, C., Construction of an infectious *Macrobrachium rosenbergii* nodavirus from cDNA clones in Sf9 cells and improved recovery of viral RNA with AZT treatment, p. 9, Copyright (2018), with permission from Elsevier; (**C**,**D**) Reprinted from Aquaculture, Vol. 319 (1–2), Hayakijkosol, O., La Fauce, K., Owens, L., Experimental infection of redclaw crayfish (*Cherax quadricarinatus*) with *Macrobrachium rosenbergii* nodavirus, the aetiological agent of white tail disease, p. 5, Copyright (2011), with permission from Elsevier).

## Data Availability

Not applicable.

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
