# Peer review of "Viral Shrimp Diseases Listed by the OIE: A Review"

_viruses, 2022, doi:10.3390/v14030585_

Round 1

Reviewer 1 Report

Dear Editor,

This manuscript “Shrimp Disease Listed in the OIE – Virus: A review” provided a comprehensive review of the information of DNA and RNA shrimp viral disease designated by the OIE and identify the latest shrimp disease trends. This paper is very useful for the research of shrimp pathogen. However, it was only showed many data (Abstract and Fig. 1-4) in the year of 2018 or 2019. For now, it was the data of 3-4 years ago. How about the data of 2020 or 2021? For a review, it is very important to summarize the latest data. So I think it has not reached the quality for published in Viruses. It was recommended to update the data and resubmit this manuscript.

Sincerely,

Mengxian Long

Author Response

Thank you for pointing this out. We agree with your comment. We made Figure (1-4) based on the FAO yearbook and FishStatJ. Unfortunately, the FAO-Fisheries and Aquaculture Statistics published on 2020 updated 2018 as the latest data, and the 2021 updated 2019 as the latest data. For that reason, we could not find data for 2020 and 2021 from FAO yearbook and FishStatJ, which lead us to conclude that the data on 2019 as the latest statistics. Therefore, we updated the data of 2019 on Figure 1-2.

Reviewer 2 Report

The manuscript is a comprehensive review of the viral crustacean diseases listed by the OIE. Although similar reviews have been published before, this manuscript presents the most detailed and comprehensive description of the diseases in question. So although the review does not offer any novel insight into or new hypotheses about viral diseases of crustaceans, it will likely work as a useful entry point into the topic and a tool for finding relevant literature about specific topics. In general the manuscript is well written, but here and there suffers from some poor language and grammatical errors. I have listed some of these places below, but there are many more throughout the manuscript.  

Specific comments:

Abstract: Poor language – please revise

All over manuscript: Penaeus and Litopenaeus (or L. and P.) is used randomly referring to the same species. Stick to one or the other (in my opinion Penaeus would be most correct).

For some diseases the molecular diagnostic methods are very detailed, whereas for others (e.g. WSSV) they are described very superficially. Maybe this could be more consistent?

Line 27: the last 40 years do not end in 2008. Why is it relevant to give that year’s production?

Line 125-126: I was not able to find anywere in reference number 20 that P. vannamei is more resistant to WSSV than P. monodon?

Line 238 and 239: “carriers” may not be the most correct term, as in many cases the animals do not get infected themselves, but merely serves as mechanical vectors. I thus believe that “vectors” would be more accurate, although I admit that there may not be complete consensus about the terminology in this issue.

Line 259: do you mean “foot bath” instead of “food bath”?

Line 334: “adult” instead of “adullt”

Line 336-338: Clarify what exactly was injected in these experiments. Was it e.g. just hemolymph or homogenate from animals carrying the non-infectious IHHNV genome sequence?

Line 341: “consists” instead of “is consist”

Line 356: “animal species” instead of “animals” and “IHHNV-susceptible” instead of “IHHNV-infected”

Line 381 - 383: Turn the sentence around – it is the animals that are susceptible not IHHNV…

Line 383: “stylirostris”

Line 384: Do you mean “tergal plates”?

Line 400: Replace toxic (with e.g. virulent or harmful or similar)

Line 476: monopartite

Line 479: Poor language – please revise

Line 491: "mass" not "mess"

Line 478 – 498: Poor language – please revise

Line 653: Nidovirales

Line 674: broodstock

Line 779: Poor language – please revise

Line 849 – 852: Poor language – please revise

Line 1008-1013: Poor language – please revise

Why is the last section called “Patents” – it does not seem to contain information about patents?

Author Response

 Comment 1: 

Abstract: Poor language – please revise

 Response 1: 

Thank you for pointing this out. We corrected all spelling and grammatical errors in the section of abstract.

“Abstract

Shrimp is not only one of the most valuable aquaculture species, but the seafood products that is traded internationally. Shrimp aquaculture has been highlighted due to the high value and raising demand, and has contributed to the economic growth in many developing countries. The production of shrimp had reached approximately 6.5 million tons in 2019 and shrimp aquaculture industry is occurring at a large scale. However, the expansion of shrimp aquaculture was accompanied by various disease outbreaks, leading to the large losses in shrimp production. Among the various diseases, viral disease had caused serious damage compared to other diseases such as bacteria and fungi. In addition, viral diseases are newly occurring, and existing diseases are evolving into new types, thereby requiring new trend information for viral diseases. Therefore, this review provides the information of on DNA and RNA shrimp viral disease designated by the OIE (World Organization for Animal Health) and identify the latest shrimp disease trends.”

 Comment 2: 

All over manuscript: Penaeus and Litopenaeus (or L. and P.) is used randomly referring to the same species. Stick to one or the other (in my opinion Penaeus would be most correct).

 Response 2: 

We agree with your opinion. We have accordingly modified Litopenaeus with Penaeus in the all over manuscript.

 Comment 3: 

For some diseases the molecular diagnostic methods are very detailed, whereas for others (e.g. WSSV) they are described very superficially. Maybe this could be more consistent?

 Response 3: 

We have supplemented the contents that need to be further developed, as the suggestion of the reviewer. We also added the reference number 229.

WSSV diagnostic technology is evolving from the past morphology-based identification technology to a highly sensitive immunological and molecular technology that can detect viruses even in asymptomatic carriers using electron microscopy (EM) [53]. Among various diagnostic methods, PCR is used as the most sensitive method for detecting WSSV infection, thereby targeting the VP28 gene [28] (Table 5). There are several PCR methods for diagnosing WSSV such as one-step PCR, nested-PCR and real-time PCR [147]. One-step PCR can be used to detect the presence of WSSV in shrimp with high levels of infection, and nested-PCR can increase the sensitivity level compared to one-step [229] can detect low levels of infection in the broodstock, nauplii, post-larvae and juvenile stages [147]. Therefore, it can be easily detected by one-step PCR when clinical signs such as lethargy, reduced feeding and white spots on the exoskeleton, but can only be detected by nested-PCR when asymptomatic [229]. In addition, real-time PCR is a reliable technique to monitor the entire analysis in actual time through detection and quantification of WSSV virion copy number. [28].”

 Comment 4: 

Line 27: the last 40 years do not end in 2008. Why is it relevant to give that year’s production?

 Response 4: 

To emphasize the growth of the shrimp aquaculture industry, we intended to explain the point at which the shrimp aquaculture production accounted for more than 50% of the global shrimp production. We revised the sentence (Line 27-30) as below: 

“The shrimp farming industry has steadily increased to about 3.4 million tons  3.6 million tons of production in the last 40 years until in 2008, accounting for more than 50% of the global shrimp market, and the main production areas for farmed shrimp are located in Southeast Asia, with China, Thailand, Vietnam, Indonesia and India being the largest producers and in the Americas, with Ecuador, Mexico and Brazil [2].”

 Comment 5: 

Line 125-126: I was not able to find anywhere in reference number 20 that P. vannamei is more resistant to WSSV than P. monodon?

 Response 5: 

Thank you for pointing this out. We revised the wrong reference and changed the sentence as below. 

Line 125-126: Due to disease problems such as WSSV, shrimp farming in Vietnam and Malaysia (2000) and India (2001) was converted from P. monodon to P. vannamei, which is relatively resistant to disease [20].

“In Asia, during the early 2000s, SPF species P. vannamei was imported from the Americas to avoid disease problems such as WSSV, resulting in the conversion of the predominant farmed species from P. monodon to P. vannamei. However, the translocation of broodstock unscreened or inadequately tested for WSSV has led to the spread of WSSV back to Asia from the Americas [12, 19].”

 Comment 6: 

Line 238 and 239: “carriers” may not be the most correct term, as in many cases the animals do not get infected themselves, but merely serves as mechanical vectors. I thus believe that “vectors” would be more accurate, although I admit that there may not be complete consensus about the terminology in this issue.

 Response 6: 

Thank you for pointing this out. We revised the contents as your suggestion.

“Aquatic and benthic organisms such as polychaete worms, microalgae and rotifer eggs are known carriers vectors of WSSV, and 43 arthropods have been reported as hosts and carriers of WSSV in culture facilities, aquatic systems or experiment [18].”

 Comment 7: 

Line 259: do you mean “foot bath” instead of “food bath”? 

 Response 7: 

Thank you for pointing this out. We revised the contents as your suggestion.

“Biosecurity measures (specific pathogen-free (SPF) broodstock, complete dry-out of culture tanks after harvest, low water exchange systems such as RAS), restricting access to vectors and pathogens (through crab fence, bird blocking, and food baths foot baths in shrimp farm entrance), and improving disease resistance (immunostimulants, neutralization, environmental management and vaccines) in shrimp, are effective, because there is no way to treat WSSV infection in shrimp [21].”

 Comment 8: 

Line 334: “adult” instead of “adullt”

 Response 8: 

Thank you for pointing this out. We revised the contents as your suggestion.

“IHHNV may exhibit different virulence due to differences in genotype of IHHNV, host susceptibility and developmental stage of infected shrimp; i) Acute infection: IHHNV-infected postlarvae and juveniles P. stylirostris sink to the bottom without swimming and can cause up to 90% of shrimp mortality in a short period of time; ii) Chronic infection: Mass mortality does not usually occur in IHHNV-infected juvenile P. vannamei and P. monodon, and sub-adults Macrobrachium rosenbergii, which can cause RDS such as growth and rostrum retardation, abdominal and tail fan deformation, cuticular roughness, and wrinkled antennal flagella, resulting in 30-90% growth retardation; iii) Asymptomatic carriers: Mytilus edulis and adullt adult M. rosenbergii can carry the infectious IHHNV type, but do not show major clinical and pathological symptoms and serve only as carriers; iv) non-infectious IHHNV insertion into shrimp host genome: Exposure to IHHNV was not infectious in P. monodon and P. vannamei individuals injected with non-infectious IHHNV through feeding and injection [57].”

 Comment 9: 

Line 336-338: Clarify what exactly was injected in these experiments. Was it e.g. just hemolymph or homogenate from animals carrying the non-infectious IHHNV genome sequence?

 Response 9: 

Thank you for pointing this out. We have clarified the contents according to your suggestion.

“Exposure to IHHNV was not infectious in P. monodon and P. vannamei individuals injected with non-infectious IHHNV crude extracts of P. monodon carrying the IHHNV sequence through feeding and injection [57].”

 Comment 10: 

Line 341: “consists” instead of “is consist”

 Response 10: 

Thank you for pointing this out. We revised the contents as your suggestion.

“The IHHNV genome is consist of consists three ORFs (open reading frames) such as two encoding nonstructural proteins (NS1; 2,001 bp and NS2; 1,092 pb) and one encoding viral capsid proteins (CP; 990 bp) [55, 57] (Table 2).”

 Comment 11:

Line 356: “animal species” instead of “animals” and “IHHNV-susceptible” instead of “IHHNV-infected”

 Response 11: 

Thank you for pointing this out. We revised the contents as your suggestion.

“IHHNV was found in P. monodon in Southeast Asia (Thailand, Taiwan, and the Philippines), and only about 30 animals species are known to be IHHNV-infected IHHNV-susceptible or carriers of IHHNV [57].”

 Comment 12:

Line 381 - 383: Turn the sentence around – it is the animals that are susceptible not IHHNV…

 Response 12: 

Thank you for pointing this out. We have clarified the contents according to your suggestion.

“IHHNV is more susceptible in postlarvae and juvenile shrimp than adults owing to more actively dividing cells in in postlarvae and juvenile shrimp.”

It is the post-larvae and juvenile shrimp that are susceptible to IHHNV owing to the reason that they have actively dividing cells.

 Comment 13:

Line 383: “stylirostris”

 Response 13: 

Thank you for pointing this out. We revised the contents as your suggestion.

The P. sytlirostris P. stylirostris presents acute symptoms of IHHNV such as white or buff-colored spots at the junction of the gergal plates in the abdomen, whereas IHHNV in the P. vannamei appears as a chronic disease, RDS, showing symptoms such as wrinkled antennal flagella, ‘bubble-heads’, deformed rostrum, cuticular roughness and deformation in 6th abdominal segment and tail fan [57].

 Comment 14:

Line 384: Do you mean “tergal plates”?

 Response 14: 

Thank you for pointing this out. We revised the contents as your suggestion.

The P. sytlirostris P. stylirostris presents acute symptoms of IHHNV such as white or buff-colored spots at the junction of the gergal plates tergal plates in the abdomen, whereas IHHNV in the P. vannamei appears as a chronic disease, RDS, showing symptoms such as wrinkled antennal flagella, ‘bubble-heads’, deformed rostrum, cuticular roughness and deformation in 6th abdominal segment and tail fan [57].

 Comment 15:

Line 400: Replace toxic (with e.g. virulent or harmful or similar)

 Response 15: 

Thank you for pointing this out. We revised the contents as your suggestion.

Post-larvae M. rosenbergii with IHHNV infection showed a high mortality rate of up to 80-100, and juvenile and subadult P. stylirostris showed a mortality rate of up to 90% (however, P. stylirostris also has increased resistance to IHHNV infection, and no significant mortality has recently been reported.); on the other hand, in P. vannamei and P. monodon, IHHNV was less toxic virulent with no death, just including RDS such as stunting and cuticular deformities [56, 64].

 Comment 16:

Line 476: monopartite

 Response 16: 

monopatite  monopartite

Thank you for pointing this out. We revised the contents as your suggestion.

IMNV is a single molecule of double stranded RNA monopatite monopartite genome of 7,561 ~ 8,230 bp in length with two open reading frames (ORFs), non-enveloped icosahedral virus with a diameter of 40 nm and fiber-like protrusions on the surface [72, 80] (Table 1).

 Comment 17:

Line 479: Poor language – please revise

 Response 17: 

We revised the contents as your suggestion. 

Totiviridae family consists of five <479> genera (1Giardiavirus, 2Leishmaniavirus and 3Trichomonasvirus infect protozoa; 4Totivirus and 5Victorivirus infect fungi) recognized by the ICTV (International Committee on Taxonomy of Viruses), but many researchers have recently suggested that the Arthropod Totiviruses should be classified separately as a "Artivirus" an Artivirus genus that is a new genus within the Totiviridae family [74].

 Comment 18:

Line 491: "mass" not "mess"

 Response 18: 

Thank you for pointing this out. We revised the contents as your suggestion.

The second half of ORF1 play a role in encoding a capsid protein with a molecular mess mass, 106 kDa [75].

 Comment 19:

Line 478 – 498: Poor language – please revise

 Response 19: 

We revised the contents as your suggestion. 

“IMNV is taxonomically a totivirus belonging to Totiviridae family that is similar to group with Protozoa and Fungal virus (In a phylogenetic analysis based on RdRp, IMNV was identified as a member of the Totiviridae family in 2008) [72, 81]. Totiviridae family consists of five genera (1Giardiavirus, 2Leishmaniavirus and 3Trichomonasvirus infect protozoa; 4Totivirus and 5Victorivirus infect fungi) recognized by the ICTV (International Committee on Taxonomy of Viruses), but many researchers have recently suggested that the Arthropod Totiviruses should be classified separately as an "Artivirus" genus that is a new genus within the Totiviridae family [74].

The whole-genome sequencing of IMNV revealed two ORFs such as ORF1, encoding RNA binding and capsid proteins and ORF2, encoding putative RNA-dependent RNA polymerase (RdRp) [81] (Table 2). The coding region of RNA-binding protein is situated in the first half of ORF 1 (including a dsRNA-binding motif). The second half of ORF1 plays a role in encoding a capsid protein with a molecular mass, 106 kDa [75]. The dsRBM (dsRNA binding motif) function as a critical in modulation or viral replication for the immune reactions in shrimp host. On the other hand, the functions of small proteins are still unclear, and just suggest hypothesis, which may be connected to assembly, cell entry and extracellular transmission of the virus [74]. The ORF2 demonstrate high similarity to the RdRp of Totiviridae family, and the ORF2 coding strategies of IMNV are similar to the strategies of GLV (Giardia lamblia virus) and other Totiviridae family, thereby suggesting the RdRp-conserved domain [74].”

 Comment 20:

Line 653: Nidovirales

 Response 20: 

Thank you for pointing this out. We revised the contents as your suggestion.

YHV is taxonomically classified in a Okavirus genus belong to Roniviridae family within Nidovirirales Nidovirales order [99] (Table 1).

 Comment 21:

Line 674: broodstock

 Response 21: 

Thank you for pointing this out. We revised the contents as your suggestion.

The mutant YHV genotype was also detected in healthy P. monodon bloodstock broodstock in Thailand and was also reported in P. monodon and P. japonicus cultured in Taiwan [94].

 Comment 22:

Line 779: Poor language – please revise

 Response 22: 

We revised the contents as your suggestion.

The name of TSV disease comes from the Taura River in Ecuador, where disease and mortality were first reported [52]. 

 Comment 23:

Line 849 – 852: Poor language – please revise

 Response 23: 

We revised the contents as your suggestion. 

Although P. vannamei is known to be the main infective as the main host for TSV, several other penaeid species (P. stylirostrisP. setiferusP. aztecusP. duorarumP. chinensis and P. monodon) have also been shown to be susceptible to identified susceptibility through experimental challenge infection. 

 Comment 24:

Line 1008-1013: Poor language – please revise

 Response 24: 

We revised the contents as your suggestion. 

Shrimp infected with MrNV targets myonuclei in the lower abdomen and haemocytes haemocytes and myonuclei in the lower abdomen and then spreads to the rest of the abdomen, and it can spread throughout the body via the hemolymph circulatory system thereby observing the almost tissues of infected shrimp except for hepatopancreas and eyestalks [139]. MrNV, a viral particle with an initial diameter of 27 nm, was observed in WTD-infected shrimp, and shortly thereafter, a second type of virus particle with an abnormally small diameter of 15 nm was observed in the WTD-infected shrimp tissue shortly thereafter, which was named XSV [149].

 Comment 25:

Why is the last section called “Patents” – it does not seem to contain information about patents?

 Response 25: 

We deleted the section of "Patents". 
